# Comparison of oral anticoagulants for stroke prevention in atrial fibrillation using the UK clinical practice research Datalink Aurum: A reference trial (ARISTOTLE) emulation study

Emma Maud Powell[1]*, Usha Gungabissoon[2], John Tazare[3], Liam Smeeth[1], Paris J. Baptiste[4], Turki M. Bin Hammad[1,5], Angel Y. S. Wong[1], Ian J. Douglas[1], Kevin Wing[1,6]

1 Department of Non-communicable Disease Epidemiology, Faculty of Epidemiology and Population Health, London School of Hygiene and Tropical Medicine, London, United Kingdom, 2 Epidemiology, GSK, London, United Kingdom, 3 Department of Medical Statistics, Faculty of Epidemiology and Population Health, London School of Hygiene and Tropical Medicine, London, United Kingdom, 4 Clinical Effectiveness Group, Centre for Primary Care, Wolfson Institute of Population Health, Barts and The London School of Medicine and Dentistry, Queen Mary University of London, London, United Kingdom, 5 Methodology and biostatistics team, Department of Efficacy and Safety, Drug sector, Saudi Food and Drug Authority, Riyadh, Saudi Arabia, 6 School of Health and Wellbeing, University of Glasgow, Glasgow, United Kingdom

* maud.teoh@lshtm.ac.uk

**Data Availability Statement:** Data are not publicly available but are available subject to protocol approval via CPRD's Research Data Governance Process (https://cprd.com/data-access) for researchers who meet the criteria for access to

## Abstract

### Background

Stroke prevention guidance for patients with atrial fibrillation (AF) uses evidence generated from randomised controlled trials (RCTs). However, applicability to patient groups excluded from trials remains unknown. Real-world patient data provide an opportunity to evaluate outcomes in a trial analogous population of direct oral anticoagulants (DOACs) users and in patients otherwise excluded from RCTs; however, there remains uncertainty on the validity of methods and suitability of the data.

Successful reference trial emulation can support the generation of evidence around treatment effects in groups excluded or underrepresented in trials.

We used linked United Kingdom primary care data to investigate whether we could emulate the pivotal ARISTOTLE trial (apixaban versus warfarin) and extend the analysis to investigate the impact of warfarin time in therapeutic range (TTR) on results.

### Methods and findings

Patients with AF in the UK Clinical Practice Research Datalink (CPRD Aurum) prescribed apixaban or warfarin from 1 January 2013 to 31 July 2019 were selected. ARISTOTLE eligibility criteria were applied to this population and matched to the RCT apixaban arm on baseline characteristics creating a trial-analogous apixaban cohort; this was propensity-score matched to warfarin users in the CPRD Aurum. ARISTOTLE outcomes were assessed using Cox proportional hazards regression stratified by prior warfarin exposure status during

confidential data. The data underlying the results presented in the study are available from CPRD (https://www.cprd.com).

**Funding:** This work was supported by the Medical Research Council (grant number MR/N013638/1 to EMP). The funders had no role in study design, data collection and analysis, decision to publish, or preparation of the manuscript.

**Competing interests:** EMP was funded by a Medical Research Council studentship for this work, and is an employee of and holds stock in Compass Pathways outside the submitted work. UG is an employee of and holds stock in GSK. JT reports no conflict of interest and is supported by an unrestricted grant from GSK. LS reports grants from Wellcome, MRC, NIHR, BHF, Diabetes UK, ESRC, EU and GSK, personal fees from GSK and AstraZeneca, and is a trustee of the British Heart Foundation, outside the submitted work. PJB is supported by Barts Charity (MGU0504) and was supported by a GSK studentship at the time of writing. TBH reports no conflict of interest. AW reports no conflict of interest and is supported by a fellowship from British Heart Foundation (FS/19/19/34175). IJD reports grants, and holds stocks in GSK, outside the submitted work. KW has nothing to disclose.

**Abbreviations:** AF, atrial fibrillation; CPRD, Clinical Practice Research Datalink; DOAC, direct oral anticoagulant; EHR, electronic health record; HES, Hospital Episodes Statistics; HR, hazard ratio; INR, international normalised ratio; IPTW, inverse probability treatment weighting; MI, myocardial infarction; NICE, National Institute for Health and Care Excellence; OAC, oral anticoagulant; ONS, Office of National Statistics; PS, propensity score; PSM, propensity score matching; RCT, randomised controlled trial; SE, systemic embolism; TIA, transient ischaemic attack; TTR, time in therapeutic range; VKA, vitamin K antagonist.

2.5 years of patient follow-up and results benchmarked against the trial results before treatment effectiveness was further evaluated based on (warfarin) TTR.

The dataset comprised 8,734 apixaban users and propensity-score matched 8,734 warfarin users. Results [hazard ratio (95% confidence interval)] confirmed apixaban noninferiority for stroke or systemic embolism (SE) [CPRD 0.98 (0.82,1.19) versus trial 0.79 (0.66,0.95)] and death from any cause [CPRD 1.03 (0.93,1.14) versus trial 0.89 (0.80,0.998)] but did not indicate apixaban superiority. Absolute event rates for stroke/SE were similar for apixaban in CPRD Aurum and ARISTOTLE (1.27%/year), whereas a lower event rate was observed for warfarin (CPRD Aurum 1.29%/year, ARISTOTLE 1.60%/year).

Analysis by TTR suggested similar effectiveness of apixaban compared with poorly controlled warfarin (TTR < 0.75) for stroke/SE [0.91 (0.73, 1.14)], all-cause death [0.94 (0.84, 1.06)], and superiority for major bleeding [0.74 (0.63, 0.86)]. However, when compared with well-controlled warfarin (TTR $\geq$ 0.75), apixaban was associated with an increased hazard for all-cause death [1.20 (1.04, 1.37)], and there was no significant benefit for major bleeding [1.08 (0.90, 1.30)]. The main limitation of the study's methodology are the risk of residual confounding, channelling bias and attrition bias in the warfarin arm, and selection bias and misclassification in the analysis by TTR.

## Conclusions

Analysis of noninterventional data generated results demonstrating noninferiority of apixaban versus warfarin consistent with prespecified benchmarking criteria. Unlike in ARISTOTLE, superiority of apixaban versus warfarin was not seen, possible due to the lower proportion of Asian patients and higher proportion of patients with well-controlled warfarin compared to ARISTOTLE. This methodological template can be used to investigate treatment effects of oral anticoagulants in patient groups excluded from or underrepresented in trials and provides a framework that can be adapted to investigate treatment effects for other conditions.

## Author summary

### Why was this study done?

- Stroke prevention treatment guidelines for patients with atrial fibrillation (AF) are based on results from randomised controlled trials (RCTs); we do not know if these results are relevant to patients that would not have been eligible to be included in the RCTs.

- This study used routinely collected health data from the United Kingdom to emulate an RCT that compared apixaban to warfarin, ARISTOTLE, and also looked at whether the benefit of apixaban compared with warfarin was impacted by the quality of warfarin therapy (measured by time in therapeutic range (TTR)).

- Emulating an RCT for stroke prevention in patients with AF should help to understand how transferable RCT results are to "real-world" practices and whether this

methodological approach can help to improve treatment options and outcomes for patient groups currently underrepresented in clinical trials.

## What did the researchers do and find?

- The researchers looked at patients with AF in a UK primary care data prescribed apixaban or warfarin and applied a "reference trial emulation" approach, in which the ARISTOTLE trial eligibility, selection, and analysis approaches were applied to UK primary care data and results benchmarked against those of ARISTOTLE.

- Patients prescribed apixaban had similar rates of outcomes to those prescribed warfarin in our cohort, and our results were successfully benchmarked against ARISTOTLE. Unlike ARISTOTLE, we did not see superiority of apixaban versus warfarin [hazard ratio (95% confidence interval)] for time to stroke or systemic embolism: 0.98 (0.82,1.19) in our cohort versus 0.79 (0.66,0.95) in ARISTOTLE.

- We also found the benefit of apixaban versus warfarin differed for some outcomes depending on the quality of warfarin therapy with apixaban (i) superior only to poorly controlled warfarin therapy for major bleeding [TTR $< 0.75$: 0.74 (0.63, 0.86), TTR $\geq 0.75$: 1.08 (0.90, 1.30)] (ii) associated with an increased risk of death compared only to well-controlled warfarin therapy [TTR $\geq 0.75$: 1.20 (1.04, 1.37), TTR $< 0.75$: 0.94 (0.84, 1.06)].

## What do these findings mean?

- Our results support the NICE guidelines on selecting treatment for stroke prevention in patients with AF and also provide reassurance on continuing warfarin in patients with high TTR.

- We can use UK primary health care data to emulate a reference trial of treatments for the prevention of stroke in AF.

- We can use the data and methods to look at how well treatments work in patients that would not have been included in RCTs such as those with multimorbidity or patient groups underrepresented in RCTs such as ethnic minority groups and older patients.

- Study limitations include the possibility of residual confounding, a risk patients doing well on warfarin were overrepresented in our cohort, a lower proportion of Asian participants in our cohort compared with ARISTOTLE, and the likelihood of residual selection bias/misclassification in the TTR analysis.

## Introduction

Atrial fibrillation (AF) is a common type of cardiac arrhythmia with an estimated prevalence of 3.3% in UK adults aged $\geq 35$ years [1]. AF is a risk factor for stroke; patients with AF have a 5-fold increased risk of stroke compared with people without AF [2], and around a quarter of all strokes are attributed to this arrhythmia [3]. In addition, increased levels of mortality,

morbidity, and disability with longer hospital stays are observed in stroke patients with AF compared with stroke patients without AF [4,5].

Pharmacological therapy recommended to reduce the risk of stroke in AF includes the use of oral anticoagulants (OACs). The introduction of direct oral anticoagulants (DOACs) for AF since 2012 in the United Kingdom provided a choice of treatment alongside the older OAC class of vitamin K antagonists (VKAs), such as warfarin, which has been available for over 60 years. The VKA OACs require regular monitoring of international normalised ratio (INR) to keep patients in the optimal therapeutic range (typically 2.0 to 3.0) in which risk of both ischemic and bleeding events are minimised [6]. A patient may require dose adjustments to stay within their INR target range. A key measure of quality of warfarin treatment is, therefore, the time in therapeutic range (TTR), which estimates the proportion of time a patient has spent with INR within optimal range. A TTR of 0.75 or greater is often considered as indicating optimal INR control and suggests a patient is spending a high proportion of their time in their INR target range.

ARISTOTLE was a pivotal randomised controlled trial (RCT) of the DOAC apixaban designed to demonstrate noninferiority compared with warfarin in the prevention of stroke or systemic embolism (SE) in patients with AF. The results demonstrated superiority of apixaban over warfarin for both prevention of stroke/SE and safety (major bleeding) [7]. Results in the European Union patient subset from the trial suggested the observed superiority of apixaban might be dependent on how well warfarin therapy was managed in the comparator group [8], an analysis that has not yet been performed outside of trial settings. In the National Institute for Health and Care Excellence (NICE) review of ARISTOTLE, several professional groups noted the TTR of warfarin users in ARISTOTLE may be lower than what is typical in UK clinical practice [9].

Treatment guidelines for DOACs are based on evidence from RCTs; however, it is unclear whether these results extend to patient groups typically excluded from trials such as those with increased bleeding risk or severe comorbidities. While there have been a number of previous studies of DOAC effectiveness using noninterventional data, there remains uncertainty on whether the data sources and methods used have fully accounted for the lack of treatment randomisation and issues such as selection bias and confounding. Comparing results from real-world studies with RCT results is challenging due to differences in patient populations, treatment adherence, and study design. However, reference trial emulation involves use of an existing named RCT to (1) inform observational study design and (2) benchmark results against, providing confidence in validity of the selected observational methods and data [10–13]. The noninterventional analysis methods can then be applied, under a set of assumptions, to reliably estimate effects in groups of patients with AF who would have been excluded from (or underrepresented in) the reference trial [14] such as patients aged >80 that were underrepresented in ARISTOTLE compared with patients with AF in UK clinical practice and patients with increased bleeding risk that were excluded by the trial eligibility criteria.

There is increasing interest in trial emulation using observation data and in the application of recent developments in pharmacoepidemiology methods involving the inclusion of prevalent users. This study used a framework that involved coarsened exact matching to select patients matching the trial population on aggregate, and sampling of prevalent users in a way that avoids selection bias and emulates the process of screening into an RCT, to construct a cohort of patients similar to the target trial population that included both new and prevalent users. This methodological approach could be adapted to a variety of treatments and different therapeutic areas.

This study sought to (1) create an ARISTOTLE-analogous cohort using routinely collected primary and secondary care data in the UK, (2) benchmark results obtained in the

ARISTOTLE-analogous cohort with ARISTOTLE results, and (3) explore whether apixaban treatment-effects in clinical practice are influenced by how well warfarin therapy is controlled.

## Materials and methods

This study is reported as per the Strengthening the Reporting of Observational Studies in Epidemiology (STROBE) guideline (S1 STROBE Checklist).

### Study design

A propensity score (PS) matched cohort study with emulation of a reference trial (ARISTOTLE).

### Setting/data sources

**UK electronic healthcare records.** This study used noninterventional data from UK Clinical Practice Research Datalink (CPRD) Aurum, a database containing anonymised data from 738 primary care practices across England (approximately 13% of the population of England with 19 million patient records and 7 million active as of September 2018 [15]). CPRD Aurum contains information on clinical diagnoses, prescribing, referrals, tests, and demographic/lifestyle factors and is representative of the population of England in geographical spread, social deprivation, age, and sex [15]. This study also used 2 additional data sources linked to CPRD Aurum: Hospital Episodes Statistics (HES) data, which contain data on patients admitted to NHS hopsitals including diagnoses, admission, and discharge, and Office of National Statistics (ONS) mortality data.

**The reference trial (ARISTOTLE).** ARISTOTLE was a randomised, double-blind trial completed in 2011, comparing apixaban with warfarin in the prevention of stroke and SE. The trial included 18,201 patients with AF and at least 1 additional risk factor for stroke. The trial was designed to test for noninferiority of apixaban compared with warfarin (noninferiorirty margin of 1.38 for the upper limit of the 95% CI of the hazard ratio (HR) for the primary outcome) and showed apixaban superiority for (1) the primary outcome of stroke or SE (HR 0.79; 95% CI 0.66, 0.95), (2) the safety endpoint of major bleeding (HR 0.69; 95% CI 0.60, 0.80), and (3) death from any cause (HR 0.89; 95% CI 0.80, 0.99). The ARISTOTLE findings led to the NICE guidelines on stroke prophylaxis in patients with AF recommending apixaban as a treatment.

ARISTOTLE eligibility criteria and summary baseline patient characteristics were used to select a cohort of patients from CPRD Aurum analogous to the ARISTOTLE participants.

The use of CPRD and ARISTOTLE are described in a previous publication [14], and the use of CPRD for this project was approved by the MHRA Independent Scientific Advisory Committee (S1 ISAC Protocol). All data used in this study were anonymised.

### Diagnostic and therapeutic codelists

All diagnostic and therapeutic codelist files used are available at https://datacompass.lshtm.ac.uk/id/eprint/3590/.

### Patient selection

**Step 1: Application of trial eligibility criteria to patients in CPRD.** We first selected HES-linked patients registered in CPRD Aurum between 1 January 2013 and 31 July 2019, who had at least 6 months between registration and the index date. ARISTOTLE recruited both new (warfarin-naïve) and prevalent (warfarin-experienced) users of warfarin with randomisation stratified on prior warfarin (or other VKA) exposure status (warfarin naïve or

experienced). To be classified as warfarin-naïve patients were required to have no evidence of exposure to warfarin or other VKA in the 5 years prior to the index date. To enable selection of a similar cohort of patients in CPRD Aurum (including both new and prevalent users of warfarin), the following process was used in determining index date:

- apixaban users

index date = first prescription of apixaban in the study period;

apixaban user classified as warfarin-naïve or warfarin-experienced at this date

- warfarin users

for new users of warfarin: index date = first prescription of warfarin in the study period;

for prevalent users of warfarin: a pool of potential index dates was selected containing all prescription dates in the study period, with index date selected at the later treatment-history sampling stage (see step 3).

ARISTOTLE eligibility criteria (Table A2 in S1 Appendix) [7] were applied giving a trial-eligible cohort for apixaban users, a trial-eligible cohort of new users of warfarin, and a pool of potential index dates (with all potential index dates kept in regardless of ARISTOTLE eligibility at this stage) for warfarin continuers (prevalent warfarin users).

**Step 2: Selection of apixaban trial-analogous patients in CPRD.** We selected a subset of the CPRD Aurum trial-eligible apixaban patients that better matched the ARISTOTLE apixaban participants based on aggregate summaries for the following key ARISTOTLE baseline characteristics:

- Age

- Sex

- Congestive heart failure or left ventricular systolic dysfunction

- Hypertension requiring treatment

- Diabetes mellitus

- Prior stroke/transient ischaemic attack (TIA)/SE

- Level of renal impairment

- Prior VKA/warfarin exposure

To characterise the baseline patient characteristics of ARISTOTLE, we used the key publication of the trial results [7], discussion of trial results by regulatory bodies [8,9,16], and publications on the trial presenting cross-tabulations on key characteristics [17,18].

An ARISTOTLE-analogous cohort of CPRD Aurum apixaban patients was then selected using a modified form of coarsened exact matching [19] (see S1 Appendix for details).

**Step 3: Matching of apixaban trial-analogous patients to warfarin trial-eligible patients in CPRD.** To emulate ARISTOTLE, which stratified randomisation on prior VKA exposure status, patients in the CPRD cohort were matched separately within the VKA-naïve and VKA-experienced strata. A 3-step procedure, based on methods proposed by Suissa and colleagues [20] and Webster-Clark and colleagues [21], was used to select and match patients in the VKA-experienced strata while avoiding selection bias; this procedure is summarised in Fig 1 and described in S1 Appendix.

The trial-analogous CPRD Aurum apixaban patients were matched to warfarin CPRD Aurum patients using greedy nearest-neighbour matching on the logit of the PS; a caliper of 0.2 times the standard deviation of the logit of the PS was used for matching as recommended by Austin [22].

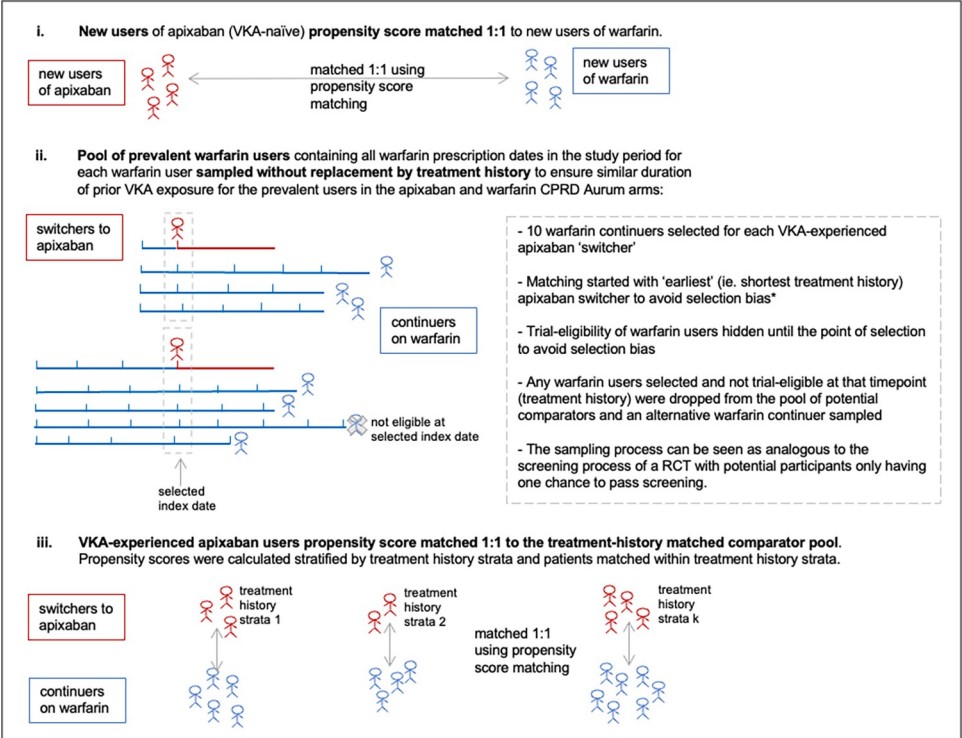

**Fig 1. Matching of apixaban trial-analogous patients to warfarin trial-eligible patients** *This method has been found in a simulation study [21] to give unbiased results.* CPRD, Clinical Practice Research Datalink; RCT, randomised controlled trial; VKA, vitamin K antagonist.

The covariates included in the PS models are detailed in Table 1.

The model resulting in the most balanced cohort was chosen with balance assessed by looking at standardised differences across all variables after matching using a target threshold of 0.05 for the maximum difference allowed for any individual variable. Balance of covariates considered to be most important in predicting outcome were prioritised, namely, age, sex, and stroke risk factors.

## Exposures and outcomes

**Exposures.**   Exposure to apixaban (5 mg/2.5 mg) or warfarin was determined using CPRD prescribing records with no restrictions on the dose prescribed.

**Outcomes.**   The primary effectiveness outcome was the composite of stroke (ischemic or haemorrhagic) or SE; individual components of this outcome (stroke, ischemic or uncertain type of stroke, haemorrhagic stroke, SE) and death from any cause were the key secondary effectiveness outcomes. Secondary effectiveness outcomes included myocardial infarction (MI), pulmonary embolism or deep vein thrombosis, and composite endpoints of effectiveness outcomes. The primary safety outcome was major bleeding (including by location—intracranial, gastrointestinal, or other location such as urinary or gynaecological). All outcomes involved hospitalisation or death and were ascertained using HES and ONS data. The ICD-10 codes used in ascertaining stroke occurrence have been recommended as having high positive predictive value [23].

**Table 1. Covariates Included in the PS models.**

| Category | Variable List |
|---|---|
| Demographics | age, sex, ethnicity |
| CHADS$_2$ stroke risk factors | congestive heart failure or left ventricular systolic dysfunction, hypertension requiring treatment, diabetes mellitus, prior stroke/TIA/SE |
| Vascular stroke risk factors | prior myocardial infarction, peripheral artery disease, aortic plaque, history of pulmonary embolism or deep vein thrombosis |
| Other risk factors | body mass index, systolic blood pressure, history of bleeding, smoking status, alcohol consumption, socioeconomic status (imd2105_5), ethnicity |
| Concomitant medications | aspirin, clopidogrel, NSAIDs, antacids, statins, ACEIs or ARBs, beta blockers, calcium channel blockers, statins, amiodarone, digoxin, proton pump inhibitors, H2 receptor antagonist |
| Comorbidities | renal function, history of fall, Charlson comorbidity components (COPD, connective tissue disease, peptic ulcer disease, liver disease, hemiplegia, cancer, haematological cancer), healthcare utilisation (number of GP consults in the prior year, number of hospitalizations in the prior year) |
| AF factors | time since AF diagnosis, history of valvular disease, history of valvular surgery |
| Healthcare utilisation | number of GP consults in the prior year, number of hospitalizations in the prior year |

ACEI, angiotensin-converting enzyme inhibitor; AF, atrial fibrillation; ARB, angiotensin receptor blocker; COPD, chronic obstructive pulmonary disease; GP, general practicioner; NSAID, nonsteroidal anti-inflammatory drug; PS, propensity score; SE, systemic embolism; TIA, transient ischemic attack.

## Statistical analysis

**Methods of analysis.**  A prospective protocol was published prior to the analysis detailing the planned analyses ([14]; also in S1 Appendix).

Changes from the planned protocol are described in Table 2.

All time-to-event endpoints were analysed using a Cox proportional hazards model, stratified by prior VKA status (experienced, naïve). The effectiveness outcomes were analysed using the intention-to-treat principle, and major bleeding was analysed using an on-treatment censoring scheme. Patients were censored at 2.5 years after index date reflecting typical maximum duration of follow-up in ARISTOTLE. Cluster-robust standard errors were used with pair membership as the clustering variable [24,25]. The proportional hazards assumption was assessed by looking at the log-log of the Kaplan–Meier survival curves and inspection of scaled Schoenfeld residuals plotted against time. Analyses were performed using SAS version 9.4 and R version 4.2.1.

**Supplementary analyses.**  A protocol planned analysis in the subset of patients deemed adherent (with adherence measured by TTR in the warfarin users and by proportion of days covered by prescriptions in the apixaban users) was planned to assess the impact of adherence on outcomes. The planned analysis was not possible due to the apixaban prescription data not providing a useful measure of adherence. An analysis by INR TTR was performed instead to assess the impact of warfarin control on results with all outcomes analysed by TTR (TTR < 0.75 and TTR ≥ 0.75). Individual predicted TTR based on baseline variables was used for patients missing TTR. In order to perform the TTR analysis while maintaining balance in the baseline covariates, inverse probability treatment weighting (IPTW) was used to rebalance the baseline characteristics, applying stabilised weights to the ARISTOTLE-analogous apixaban users. A similar approach to the main analysis was used with PS models constructed separately for the new users and warfarin-experienced users.

An additional post hoc analysis was performed looking at the proportion of apixaban patients prescribed reduced-dose apixaban along with a comparison of the patients meeting

**Table 2. Changes from planned analyses.**

| Original Planned Analysis | Updated Analysis | Reason for Change |
|---|---|---|
| Patients to be selected from both CPRD GOLD and CPRD Aurum. | Only CPRD Aurum used. | There was a much larger sample size available in CPRD Aurum meaning combining of the 2 data sources was not required. |
| Censoring scheme to censor at 5 years after index date. | Censoring scheme censored at 2.5 years after index date. | The ARISTOTLE trial had median duration of follow-up of 1.8 years (IQR 1.4, 2.3); therefore, a 2.5-year cutoff gives a more similar duration of follow-up than 5 years. |
| Adherence of apixaban users to be measured by proportion of days covered by prescriptions. | Treatment persistence measured instead (proportion of patients still on index treatment at date of censoring). | Repeat prescriptions are often issued automatically, meaning comparing number of days covered by prescribed pills to the number of days in the treatment period did not provide useful insight on adherence. |
| Supplementary analysis in patients deemed adherent (PDC $\geq$ 80%, ARISTOTLE compliance limit). | Analysis by TTR only. | Unable to ascertain useful measure of adherence in the apixaban users. |
| Noninferiority will be concluded when the upper limit of the 95% CI for the HR must be less than 1.52 (upper limit in the EU subgroup of ARISTOTLE). | Noninferiority will be concluded when the upper limit of the 95% CI for the HR is less than 1.38 (same noninferiority margin of ARISTOTLE). | The noninferiority margin used in ARISTOTLE was the one agreed by regulators to represent the maximum acceptable clinical difference. By applying the same margin, we ensure that the conclusion is based on more rigorous criteria. |
| Aim to include prior INR control in propensity model for VKA-experienced patients. | Primary analysis does not include prior INR control. Post hoc sensitivity analysis performed including prior INR control in the PS model. | High rate of missing data for prior INR control made it not advisable to include this variable in the PS model for the main analysis. Other variables predictive of poor INR control such as age are already included. Post hoc sensitivity analysis including INR control in the PS model performed to assess the potential impact of not including this variable following question in peer review on the omission of this variable. |
| N/A | Post hoc analysis assessing apixaban dose-adjustment in CPRD Aurum | Suggested by peer review to provide evidence on the quality of dose adjustment in CPRD Aurum and how this may impact the results in the trial-analogous cohort. |

CPRD, Clinical Practice Research Datalink; HR, hazard ratio; INR, international normalised ratio; IQR, interquartile range; PS, propensity score; TTR, time in therapeutic range; VKA, vitamin K antagonist.

the criteria for dose-reduction against the dose actually prescribed. In this analysis, apixaban dose in the ARISTOTLE-analogous CPRD cohort was assessed and compared against the ARISTOTLE protocol-specified criteria and NICE criteria for reduced apixaban dose. ARISTOTLE specified that participants meeting any 2 of the following criteria assessed at the time of randomisation should have their apixaban dose reduced to 2.5 mg BID: age $\geq$80 years, body weight $\leq$60 kg, or serum creatinine $\geq$1.5 mg/dL. These criteria are equivalent to the NICE guidelines for dose reduction with NICE having an additional criteria indicating reduced dose in those with creatinine clearance 15 to 29 mL/minute.

In addition, to assess the impact of the quality of dose-adjustment in the CPRD cohort on the observed effectiveness of apixaban relative to warfarin, a supplementary post hoc analysis was performed looking at the results in the subset of apixaban patients prescribed the correct dose compared with IPTW rebalanced warfarin comparators.

**Sensitivity analyses.** Primary and secondary effectiveness outcomes were also analysed using the on-treatment censoring scheme to investigate whether treatment discontinuation compromises confidence in the effectiveness analyses.

Treatment persistence was defined by looking at longitudinal prescription data for OACs; OAC treatment windows were derived in which gaps $> = 6$ months between prescription dates were considered as distinct treatment windows. The end of each OAC treatment window was derived as the date of the last prescription of index OAC + the number of days supply given in the last prescription + a grace period of 30 days. In cases of overlapping OAC

treatment windows, the date of the first prescription of the subsequent OAC treatment window was used to define the end of the prior OAC window. A prescription for a different OAC from the index OAC treatment was considered as a treatment switch. An ending of index OAC treatment with no subsequent prescription for any other OAC recorded was considered as treatment stop. Gaps of > = 6 months with no subsequent OAC prescriptions recorded were categorised as having stopped OAC treatment.

The set of patients who switched or discontinued treatment during follow-up were examined to ascertain whether selection bias due to attrition may have affected the on-treatment analyses (Table A9 in S1 Appendix).

Apixaban was first launched for AF in the UK in January 2013, with relatively few patients receiving a prescription in the first year it was available; we therefore performed a sensitivity analysis with the start of study period shifted forwards a year to investigate the impact of inclusion of early adopters who may differ from later adopters of a new drug.

**Confounding and bias.**   In the study period, apixaban was a newly available treatment leading to the possibility of channelling bias [26]. By applying trial eligibility criteria to both treatment cohorts and matching using baseline covariates, we aimed to minimise channelling bias. To handle confounding, treatment arms were matched using propensity score matching (PSM) [27].

**Benchmarking results against ARISTOTLE.**   The study hypothesis was that results in the CPRD ARISTOTLE-analogous cohort would be comparable to the ARISTOTLE results, as defined by the prespecified benchmarking criteria. A slightly weaker benefit of apixaban versus warfarin was expected based on the weaker benefit seen in the EU subgroup of ARISTOTLE, and an expectation that the quality of warfarin control in UK patients may be higher than that observed in ARISTOTLE.

The benchmarking criteria for considering the results in the trial-analogous CPRD cohort to be comparable with ARISTOTLE were prespecified and published previously [14]:

- The effect size must be clinically comparable with the ARISTOTLE findings; the HR for time to stroke/SE with the HR must be between 0.69 and 0.99. This range is not symmetrical around the ARISTOTLE estimate of 0.79 as it is anticipated the treatment effect in routine clinical care may be weaker than that seen in the optimised setting of a clinical trial.

- The upper limit of the 95% CI for the HR for time to stroke/SE must be less than 1.38 (non-inferiority margin used in ARISTOTLE, updated since protocol—see Table 2).

The benchmarking step applied only to the primary effectiveness outcome in the trial-analogous CPRD cohort; results in other groups such as patients underrepresented or excluded from the trial would not necessarily be expected to remain consistent to the RCT results, given the relative risks may differ in these groups. Comparability of other outcomes was to be assessed descriptively with no formal criteria or hypothesis testing used.

## Missing data

Patients with missing systolic blood pressure (0.1%), body mass index (3.3%), smoking status (0.1%), or socioeconomic status (0.1%) were excluded from the trial-eligible cohort as the proportion of patients with these missing was low. Patients with missing renal function (1.3%), ethnicity (0.4%), or alcohol use (5.6%) were kept in the cohort through a missing indicator approach; this approach is valid under the assumption that these variables act as confounders and influence clinician prescribing decisions only when observed [28]. A total of 1,176 (13.3%) warfarin users in the CPRD cohort did not have INR measurements in the data during their

treatment period with predicted TTR used for these patients in the analysis by TTR (see S1 Appendix for details).

### Ethics

Scientific approval was provided by the London School of Hygiene and Tropical Medicine research ethics committee (ref 17682) and the independent scientific advisory committee of the Medicines and Healthcare Products Regulatory Agency (protocol no. 19_066R). CPRD data are already approved via a national research ethics committee for purely noninterventional research of this type. CPRD data are analysed anonmymously; therefore, individual patient consent is not sought by contributing medical practices when data are shared with CPRD; however, patients are able to opt out of their patient information being shared for research.

## Results

### Participants

Between 1 January 2013 and 31 July 2019, there were 86,888 people with AF prescribed apixaban and 159,632 prescribed warfarin in HES-linked CPRD Aurum practices (Fig 2). Application of minimum registration period and ARISTOTLE inclusion criteria reduced this to 67,539 apixaban and 139,527 warfarin patients. After applying ARISTOTLE exclusion criteria, there were 41,487 apixaban and 101,159 warfarin patients.

Selecting apixaban patients to match ARISTOTLE on key baseline characteristics yielded 9,120 apixaban patients (3,912 new users and 5,208 prevalent users) available for PSM to 101,159 warfarin patients. For 274 apixaban patients, no match could be found giving a PS matched cohort of 8,846 apixaban and 8,846 warfarin patients.

### Application of ARISTOTLE inclusion/exclusion criteria and matching to ARISTOTLE

Applying the ARISTOTLE inclusion/exclusion criteria and matching to ARISTOTLE baseline patient characteristics resulted in a cohort similar to the ARISTOTLE apixaban participants (Table 3); for example, median age was 78 and mean CHADS$_2$ score 2.4 in CPRD Aurum before applying trial criteria and matching, whereas the median age of 71 and mean CHADS$_2$ score 2.1 after these steps matched the ARISTOTLE apixaban participants. The ARISTOTLE-analogous apixaban arm matched the trial arm on prior VKA exposure, age, sex, stroke risk factors and CHADS$_2$ score, and proportion of patients with moderate or severe renal impairment.

Differences remained on baseline characteristics it was not feasible to match on, namely, ethnicity (95.2% white, 2.4% Asian in CPRD Aurum apixaban versus 82.6% white, 14.4% Asian in ARISTOTLE) and concomitant medications (amiodarone 3.8%, aspirin 5.8%, digoxin 13.9% in CPRD Aurum apixaban users versus amiodarone 11.1%, aspirin 31.3%, digoxin 32.0% in ARISTOTLE apixaban arm). See S1 Appendix for details on matching feasibility.

### Propensity score matching of CPRD Aurum trial-analogous apixaban users to CPRD Aurum warfarin users

**Results of propensity score matching.**   Before PSM, differences between treatment groups were evident for most baseline variables including age (median age 71 in apixaban versus 78 in warfarin), sex (apixaban 35.6% female versus warfarin 43.6%), and stroke risk factors (see Table 3). After PSM, all baseline characteristics were well balanced (maximum standardised difference .031). From 9,120 apixaban users, only 274 (3.0%) were dropped due to unsuccessful matching.

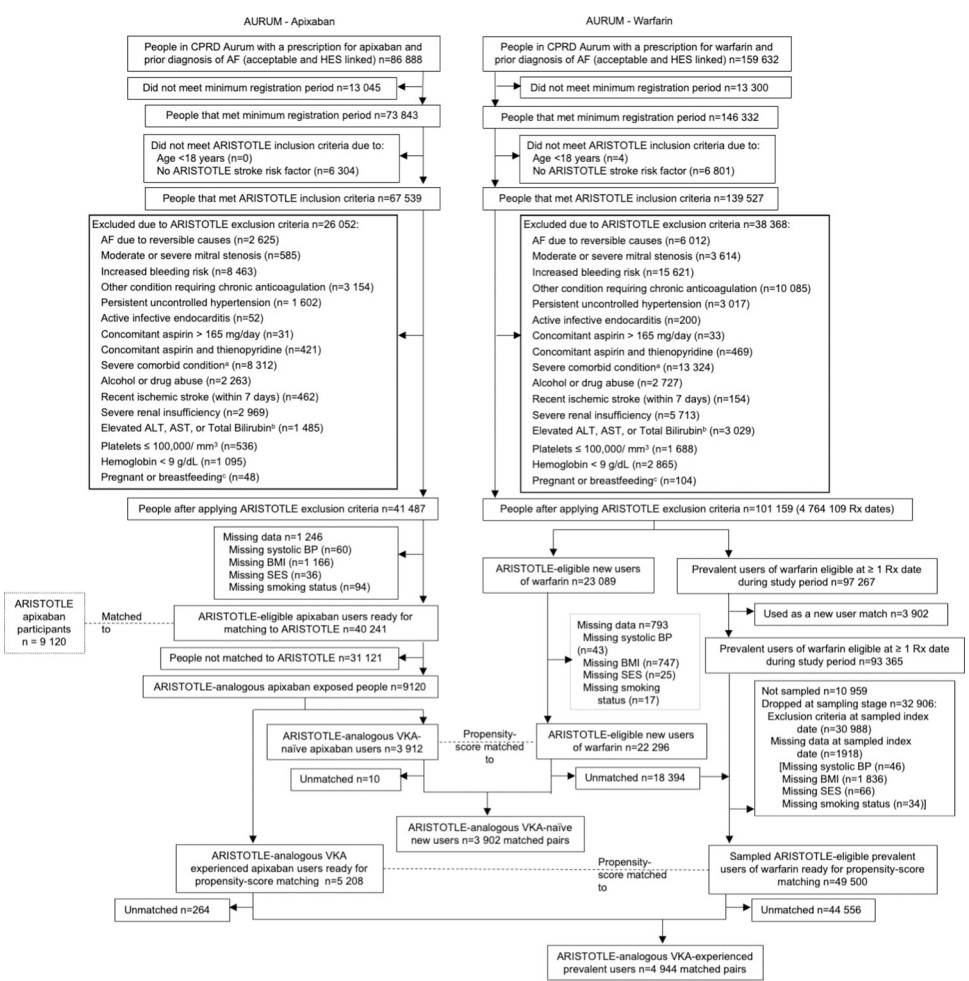

**Fig 2. Selection of ARISTOTLE-analogous CPRD Aurum Cohort.** Flow of number of individuals included in the analysis. AF, atrial fibrillation; ALT, alanine transaminase; AST, aspartate transaminase; BMI, body mass index; BP, blood pressure; CPRD, Clinical Practice Research Datalink; HES, Hospital Episodes Statistics; Rx, Prescription; SES, socioeconomic status; ULN, upper limit of normal; VKA, vitamin K antagonist. [a]Severe comorbid condition with life expectancy <1 year or reasons making participation impractical; [b]ALT or AST > 2X ULN or Total Bilirubin ≥ 1.5X ULN; [c]Pregnant or breastfeeding within 3 years prior. See Table A1 in S1 Appendix for detailed list of inclusion and exclusion criteria. Note: For prevalent warfarin users, trial eligibility only revealed at point of random selection into the cohort for prevalent users. Numbers in figure show maximum theoretical number of warfarin users available should they be selected only at a time they were eligible for the trial.

## Main results

The HR for stroke/SE in the PS matched groups was 0.98 (95% CI 0.82,1.19) (Fig 3 and Table A3 in S1 Appendix). This association was consistent with the noninferiority margin (upper limit of the 95% CI less than 1.38) [7] but did not show superiority as predicted by ARISTOTLE [HR 0.79 (95% CI 0.66,0.95)] (Fig 3 and Table A2 in S1 Appendix). The outcome of all-cause mortality also showed noninferiority [Aurum 1.03 (0.93,1.14) versus trial 0.89 (0.80,0.998)] but did not indicate apixaban superiority. Absolute event rates for the primary outcome and components were close to the trial for apixaban—for example [comparing Aurum versus trial], stroke/SE event rate of 1.27%/year versus 1.27%, whereas the warfarin group had a lower event rate compared with ARISTOTLE (stroke/SE event rate of 1.29%/year versus 1.60% and hemorrhagic stroke 0.33%/year versus 0.47%/year) (Fig 3). Mean duration of follow-up in the cohort was 1.8 years in the apixaban arm and 2.2 years in the warfarin arm.

**Table 3. Baseline characteristics of patients with AF prescribed apixaban and warfarin in CPRD Aurum compared with ARISTOTLE participants: (i) before and (ii) after applying ARISTOTLE inclusion and exclusion criteria and (iii) after matching to the trial participants.**

| | CPRD Aurum | | | | | | | ARISTOTLE Trial | |
| | No ARISTOTLE criteria or matching | | After applying ARISTOTLE criteria | | After applying ARISTOTLE criteria, matching to the trial and PSM apixaban to warfarin | | | | |
| Characteristic—n(%) unless specified | Apixaban ($N = 73,843$) | Warfarin ($N = 146,332$) | Apixaban ($N = 41,487$) | Warfarin ($N = 101,159$) | Apixaban ($N = 8,846$) | Warfarin ($N = 8,846$) | Standardised difference | Apixaban ($N = 9,120$) | Warfarin ($N = 9,081$) |
|---|---|---|---|---|---|---|---|---|---|
| Age–years, median (IQR) | 78 (70, 85) | 78 (71, 84) | 78 (71, 84) | 78 (72, 84) | 71 (63, 77) | 71 (63, 77) | 0.008 | 70 (63, 76) | 70 (63, 76) |
| Female sex | 34,430 (46.6) | 63,321 (43.3) | 19,591 (47.2) | 44,197 (43.7) | 3,144 (35.5) | 3,190 (36.1) | 0.011 | 3,234 (35.5) | 3,182 (35.0) |
| Systolic blood pressure–mm Hg, median (IQR) | 130 (120, 140) | 130 (120, 140) | 131 (120, 140) | 130 (120, 140) | 130 (120, 140) | 130 (120, 140) | 0.001 | 130 (120, 140) | 130 (120, 140) |
| Missing | 132 | 267 | 60 | 125 | 0 | 0 | | | |
| Weight–kg, median (IQR) | 79 (67, 92) | 80 (68, 93) | 80 (68, 93) | 80 (69, 94) | 85 (73, 100) | 85 (74, 99) | 0.003 | 82 (70, 96) | 82 (70, 95) |
| Prior MI | 9,958 (13.5) | 20,406 (13.9) | 5,035 (12.1) | 13,446 (13.3) | 1,090 (12.3) | 1,074 (12.1) | 0.006 | 1,319 (14.5) | 1,266 (13.9) |
| Prior clinically relevant or spontaneous bleeding | 16,972 (23.0) | 31,034 (21.2) | 7,721 (18.6) | 19,007 (18.8) | 1,533 (17.3) | 1,507 (17.0) | 0.008 | 1,525 (16.7) | 1,515 (16.7) |
| History of fall within previous year | 2,443 (3.3) | 2,688 (1.8) | 1,093 (2.6) | 1,561 (1.5) | 137 (1.5) | 131 (1.5) | 0.006 | 386 (4.2) | 367 (4.0) |
| Prior use VKA >30 days | 24,240 (32.8) | 102,725 (70.2) | 12,558 (30.3) | 75,787 (74.9) | 4,944 (55.9) | 4,944 (55.9) | 0.000 | 5,208 (57.1) | 5,193 (57.2) |
| Qualifying risk factors | | | | | | | | | |
| Age ≥75 years | 45,762 (62.0) | 93,436 (63.9) | 26,730 (64.4) | 68,197 (67.4) | 2,770 (31.3) | 2,740 (31.0) | 0.007 | 2,850 (31.2) | 2,828 (31.1) |
| Prior stroke, TIA, or SE | 20,713 (28.1) | 38,132 (26.1) | 11,422 (27.5) | 25,898 (25.6) | 1,711 (19.3) | 1,709 (19.3) | 0.001 | 1,748 (19.2) | 1,790 (19.7) |
| Heart failure or reduced LVEF | 22,329 (30.2) | 50,480 (34.5) | 11,650 (28.1) | 33,422 (33.0) | 3,052 (34.5) | 3,022 (34.2) | 0.007 | 3,235 (35.5) | 3,216 (35.4) |
| Diabetes | 20,104 (27.2) | 40,103 (27.4) | 11,630 (28.0) | 28,496 (28.2) | 2,243 (25.4) | 2,275 (25.7) | 0.008 | 2,284 (25.0) | 2,263 (24.9) |
| Hypertension req. treatment | 52,406 (71.0) | 105,097 (71.8) | 31,780 (76.6) | 76,923 (76.0) | 7,662 (86.6) | 7,669 (86.7) | 0.002 | 7,962 (87.3) | 7,954 (87.6) |
| $CHADS_2$ score, mean ± SD | 2.4 ± 1.5 | 2.4 ± 1.4 | 2.5 ± 1.3 | 2.5 ± 1.2 | 2.1 ± 1.1 | 2.1 ± 1.1 | 0.003 | 2.1 ± 1.1 | 2.1 ± 1.1 |
| $CHADS_2 = 0$ | 6,494 (8.8) | 10,240 (7.0) | 134 (0.3) | 356 (0.4) | 52 (0.6) | 55 (0.6) | 0.004 | 54 (0.6) | 58 (0.6) |
| $CHADS_2 = 1$ | 14,860 (20.1) | 28,124 (19.2) | 10,602 (25.6) | 23,539 (23.3) | 2,971 (33.6) | 2,912 (32.9) | 0.014 | 3,046 (33.4) | 3,025 (33.3) |
| $CHADS_2 = 2$ | 19,844 (26.9) | 43,294 (29.6) | 12,969 (31.3) | 32,980 (32.6) | 3,157 (35.7) | 3,239 (36.6) | 0.019 | 3,262 (35.8) | 3,254 (35.8) |
| $CHADS_2 \geq 3$ | 32,645 (44.2) | 64,674 (44.2) | 17,783 (42.9) | 44,284 (43.8) | 2,666 (30.1) | 2,640 (29.8) | 0.006 | 2,758 (30.2) | 2,744 (30.2) |
| Medications at index date | | | | | | | | | |
| ACE inhibitor or ARB | 34,899 (47.3) | 82,841 (56.6) | 21,656 (52.2) | 61,435 (60.7) | 5,529 (62.5) | 5,573 (63.0) | 0.010 | 6,464 (70.9) | 6,368 (70.1) |
| Amiodarone | 1,903 (2.6) | 4,859 (3.3) | 961 (2.3) | 3,259 (3.2) | 336 (3.8) | 322 (3.6) | 0.008 | 1,009 (11.1) | 1,042 (11.5) |
| Beta-blocker | 46,173 (62.5) | 88,274 (60.3) | 25,990 (62.6) | 62,016 (61.3) | 6,083 (68.8) | 6,031 (68.2) | 0.013 | 5,797 (63.6) | 5,685 (62.6) |
| Aspirin | 5,209 (7.1%) | 10,833 (7.4%) | 2,612 (6.3) | 6,429 (6.4) | 514 (5.8) | 557 (6.3) | 0.020 | 2,859 (31.3) | 2,773 (30.5) |
| Clopidogrel | 2,697 (3.7%) | 3,697 (2.5%) | 1,238 (3.0) | 2,177 (2.2) | 229 (2.6) | 215 (2.4) | 0.010 | 170 (1.9) | 168 (1.9) |
| Digoxin | 9,771 (13.2) | 33,342 (22.8) | 5,147 (12.4) | 23,322 (23.1) | 1,232 (13.9) | 1,244 (14.1) | 0.004 | 2,916 (32.0) | 2,912 (32.1) |
| Calcium channel blocker | 19,659 (26.6) | 39,909 (27.3) | 12,522 (30.2) | 30,379 (30.0) | 2,965 (33.5) | 2,994 (33.8) | 0.007 | 2,744 (30.1) | 2,823 (31.1) |
| Statin | 39,027 (52.9) | 82,086 (56.1) | 23,035 (55.5) | 58,647 (58.0) | 5,230 (59.1) | 5,228 (59.1) | 0.000 | 4,104 (45.0) | 4,095 (45.1) |
| Nonsteroidal anti-inflammatory | 4,953 (6.7) | 8,107 (5.5) | 2,939 (7.1) | 5,891 (5.8) | 487 (5.5) | 479 (5.4) | 0.004 | 752 (8.2) | 768 (8.5) |
| Gastric antacid drugs | 1,833 (2.5) | 3,290 (2.2) | 1,042 (2.5) | 2,346 (2.3) | 180 (2.0) | 180 (2.0) | 0.000 | 1,683 (18.5) | 1,667 (18.4) |
| Proton pump inhibitor | 2,844 (38.0) | 47,838 (32.7) | 15,197 (36.6) | 31,769 (31.4) | 3,052 (34.5) | 3,104 (35.1) | 0.012 | | |
| H2 receptor antagonist | 3,188 (4.3) | 4,837 (3.3) | 1,586 (3.8) | 3,006 (3.0) | 281 (3.2) | 250 (2.8) | 0.021 | | |
| Renal function, creatinine clearance | | | | | | | | | |
| Normal, >80 ml/minute | 21,591 (29.2) | 45,793 (31.3) | 12,261 (29.6) | 31,451 (31.1) | 4,098 (46.3) | 4,074 (46.1) | 0.005 | 3,761 (41.2) | 3,757 (41.4) |

*(Continued)*

**Table 3.** (Continued)

| | CPRD Aurum | | | | | | | ARISTOTLE Trial | |
|---|---|---|---|---|---|---|---|---|---|
| **Mild imp., >50 to 80 ml/minute** | 28,976 (39.2) | 56,742 (38.8) | 17,494 (42.2) | 41,290 (40.8) | 3,307 (37.4) | 3,292 (37.2) | 0.004 | 3,817 (41.9) | 3,770 (41.5) |
| **Moderate imp. (>30 to 50 ml/minute)** | 17,007 (23.0) | 32,881 (22.5) | 9,708 (23.4) | 23,316 (23.0) | 1,276 (14.4) | 1,306 (14.8) | 0.010 | 1,365 (15.0) | 1,382 (15.2) |
| **Severe imp. (≤30 ml/minute)** | 4,317 (5.8) | 9,251 (6.3) | 1,053 (2.5) | 4,251 (4.2) | 126 (1.4) | 132 (1.5) | 0.006 | 137 (1.5) | 133 (1.5) |
| **Not reported** | 1,952 (2.6) | 1,665 (1.1) | 972 (2.3) | 851 (0.8) | 39 (0.4) | 42 (0.5) | 0.005 | 40 (0.4) | 39 (0.4) |
| **Peripheral artery disease** | 5,984 (8.1) | 12,764 (8.7) | 2,770 (6.7) | 7,516 (7.4) | 552 (6.2) | 538 (6.1) | 0.007 | | |
| **Aortic plaque** | 17,919 (24.3) | 40,415 (27.6) | 8,974 (21.6) | 25,193 (24.9) | 2,097 (23.7) | 2,057 (23.3) | 0.011 | | |
| **Smoking status** | | | | | | | | | |
| **Nonsmoker** | 27,568 (37.3) | 51,612 (35.3) | 15,949 (38.4) | 36,338 (35.9) | 3,186 (36.0) | 3,164 (35.8) | 0.005 | | |
| **Ex-smoker** | 40,815 (55.3) | 84,850 (58.0) | 22,757 (54.9) | 58,669 (58.0) | 4,925 (55.7) | 4,945 (55.9) | 0.005 | | |
| **Current smoker** | 5,236 (7.1) | 9,658 (6.6) | 2,688 (6.5) | 6,049 (6.0) | 735 (8.3) | 737 (8.3) | 0.001 | | |
| **Missing** | 224 | 211 | 94 | 102 | 0 | 0 | | | |
| **Alcohol consumption** | | | | | | | | | |
| **Nondrinker** | 27,185 (36.8) | 52,744 (36.0) | 14,957 (36.1) | 35,905 (35.5) | 2,802 (31.7) | 2,842 (32.1) | 0.010 | | |
| **Light, 1 to 14 units per week** | 32,190 (43.6) | 66,072 (45.2) | 18,762 (45.2) | 46,876 (46.3) | 4,135 (46.7) | 4,153 (46.9) | 0.004 | | |
| **Moderate, 15 to 42 units per week** | 8,950 (12.1) | 15,916 (10.9) | 5,053 (12.2) | 11,109 (11.0) | 1,563 (17.7) | 1,515 (17.1) | 0.014 | | |
| **Heavy, >42 units per week** | 1,488 (2.0) | 2,028 (1.4) | 617 (1.5) | 1,149 (1.1) | 203 (2.3) | 204 (2.3) | 0.001 | | |
| **Missing** | 3,901 | 9,223 | 2,032 | 5,893 | 143 | 132 | | | |
| **Socioeconomic status (England IMD2015 Quintile)** | | | | | | | | | |
| **1 (least deprived)** | 18,893 (25.6) | 36,046 (24.6) | 10,867 (26.2) | 25,270 (25.0) | 2,246 (25.4) | 2,231 (25.2) | 0.004 | | |
| **2** | 17,203 (23.3) | 33,585 (23.0) | 9,768 (23.5) | 23,473 (23.2) | 2,098 (23.7) | 2,057 (23.3) | 0.011 | | |
| **3** | 14,591 (19.8) | 29,856 (20.4) | 8,207 (19.8) | 20,704 (20.5) | 1,715 (19.4) | 1,759 (19.9) | 0.013 | | |
| **4** | 12,283 (16.6) | 25,614 (17.5) | 6,767 (16.3) | 17,498 (17.3) | 1,443 (16.3) | 1,465 (16.6) | 0.007 | | |
| **5 (most deprived)** | 10,804 (14.6) | 21,066 (14.4) | 5,843 (14.1) | 14,098 (13.9) | 1,344 (15.2) | 1,334 (15.1) | 0.003 | | |
| **Missing** | 69 | 165 | 36 | 116 | 0 | 0 | | | |
| **Ethnicity** | | | | | | | | | |
| **White** | 70,703 (95.7) | 141,019 (96.4) | 39,685 (95.7) | 97,735 (96.6) | 8,424 (95.2) | 8,444 (95.5) | 0.011 | 7,536 (82.6) | 7,493 (82.5) |
| **Black** | 714 (1.0) | 1,326 (0.9) | 372 (0.9) | 821 (0.8) | 104 (1.2) | 103 (1.2) | 0.001 | 125 (1.4) | 102 (1.1) |
| **Asian** | 1,371 (1.9) | 2,481 (1.7) | 774 (1.9) | 1,536 (1.5) | 214 (2.4) | 209 (2.4) | 0.000 | 1,310 (14.4) | 1,332 (14.7) |
| **Other** | 198 (0.3) | 356 (0.2) | 113 (0.3) | 232 (0.2) | 22 (0.2) | 22 (0.2) | 0.000 | 149 (1.6) | 153 (1.7) |
| **Mixed** | 152 (0.2) | 308 (0.2) | 75 (0.2) | 190 (0.2) | 25 (0.3) | 28 (0.3) | 0.006 | 0 | 0 |
| **Unknown** | 385 (0.5) | 448 (0.3) | 252 (0.6) | 350 (0.3) | 42 (0.5) | 25 (0.3) | 0.031 | 0 | 0 |
| **Charlson comorbidity components** | | | | | | | | | |
| **Chronic obstructive pulmonary disease** | 10,324 (14.0) | 19,033 (13.0) | 5,411 (13.0) | 12,573 (12.4) | 1,138 (12.9) | 1,141 (12.9) | 0.001 | | |
| **Connective tissue disease** | 5,377 (7.3) | 9,784 (6.7) | 3,000 (7.2) | 6,744 (6.7) | 536 (6.1) | 534 (6.0) | 0.001 | | |
| **Peptic ulcer** | 4,400 (6.0) | 8,399 (5.7) | 2,161 (5.2) | 5,458 (5.4) | 411 (4.6) | 393 (4.4) | 0.010 | | |
| **Liver disease** | 761 (1.0) | 1,291 (0.9) | 263 (0.6) | 642 (0.6) | 76 (0.9) | 61 (0.7) | 0.019 | | |
| **Hemiplegia** | 265 (0.4) | 559 (0.4) | 147 (0.4) | 379 (0.4) | 24 (0.3) | 16 (0.2) | 0.019 | | |
| **Nonhaematological cancer** | 12,567 (17.0) | 23,383 (16.0) | 6,019 (14.5) | 14,413 (14.2) | 1,066 (12.1) | 1,146 (13.0) | 0.027 | | |

*(Continued)*

**Table 3.** (Continued)

| | CPRD Aurum | | | | | | | ARISTOTLE Trial | |
|---|---|---|---|---|---|---|---|---|---|
| Haematological cancer | 1,966 (2.7) | 3,481 (2.4) | 951 (2.3) | 2,231 (2.2) | 174 (2.0) | 163 (1.8) | 0.009 | | |
| BMI—kg/m², median (IQR) | 28 (24, 32) | 28 (23, 32) | 28 (25, 32) | 28 (25, 32) | 29 (26, 33) | 29 (26, 33) | 0.003 | | |
| Missing | 2 270 | 5 858 | 1 166 | 3 593 | 0 | 0 | | | |

ACE, angiotensin-converting enzyme; ARB, angiotensin-receptor blocker; BMI, body mass index; CHADS₂, stroke risk factor score based on congestive heart failure, hypertension, age ≥75 years, diabetes, prior stroke; CPRD, Clinical Practice Research Datalink; IMD2015, Index of Multiple Deprivation 2015; imp., impairment; IQR, interquartile range; LVEF, left ventricular ejection fraction; MI, myocardial infarction; PSM, propensity score matching; SD, standard deviation; SE, systemic embolism; TIA, transient ischemic attack; VKA, vitamin K antagonist.

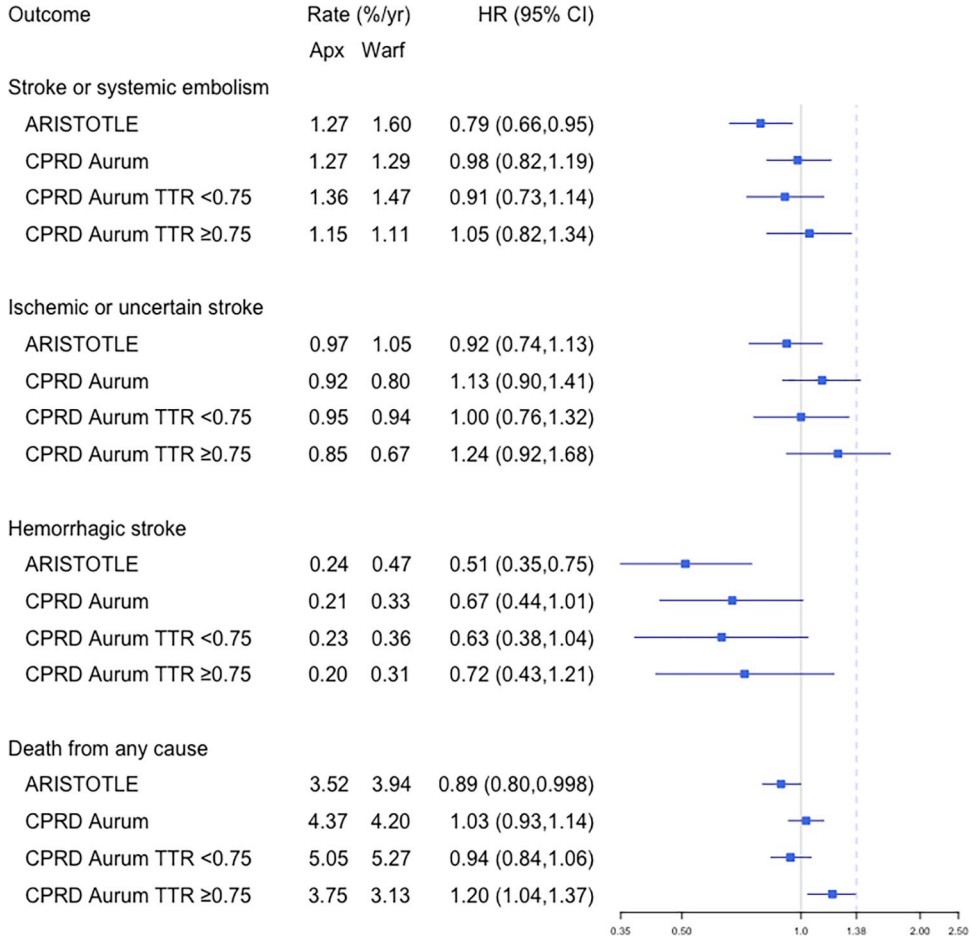

**Fig 3. Forest plot showing HRs (dots) and 95% CIs (lines) for apixaban vs. warfarin.** Absolute event rates (%/year) and HR (95% CIs) are presented for key effectiveness outcomes in (i) ARISTOTLE, (ii) CPRD Aurum trial-matched cohort, (iii) CPRD Aurum trial-matched with TTR < 0.75, and (iv) CPRD Aurum trial-matched with TTR ≥ 0.75. Dashed line shows noninferiority margin 1.38 for the upper bound of the 95% CI of the HR used in ARISTOTLE for the primary outcome of stroke or SE. For the analysis by TTR, IPTW was applied to the apixaban users targeting the treatment effect in the warfarin users with TTR < 0.75 and TTR ≥ 0.75. CI, confidence interval; CPRD, Clinical Practice Research Datalink; HR, hazard ratio; IPTW, inverse probability treatment weighting; SE, systemic embolism; TTR, time in therapeutic range.

**Table 4. Apixaban dose-adjustment in CPRD Aurum compared with ARISTOTLE.**

| | CPRD Aurum ARISTOTLE-analogous Apixaban (N = 8,846) | CPRD Aurum ARISTOTLE-analogous Warfarin (N = 8,846) | ARISTOTLE RCT Apixaban (N = 9,120) |
|---|---|---|---|
| Standard 5.0 mg BID dose | 7,580 (85.7%) | N/A | 8,692 (95.3%) |
| Reduced 2.5 mg BID dose | 1,266 (14.3%) | N/A | 428 (4.7%) |
| Reduced dose indicated per ARISTOTLE criteria | 434 (4.9%) | 436 (4.9%) | 428 (4.7%) |
| Reduced dose indicated per NICE criteria | 454 (5.1%) | 459 (5.2%) | NR |

NICE criteria for dose-adjustment included additional criteria of creatinine clearance 15–29 mL/minute.

CPRD, Clinical Practice Research Datalink; N/A, not applicable; NICE, National Institute for Health and Care Excellence; NR, not reported; RCT, randomised controlled trial.

## Analysis of impact of warfarin time in therapeutic range (TTR)

TTR was higher in the CPRD cohort than in ARISTOTLE (mean 0.73 versus 0.62, median 0.76 versus 0.66).

Analysis by TTR suggested noninferiority of apixaban versus warfarin in those with TTR < 0.75 [stroke/SE 0.91 (0.73,1.14), all-cause death 0.94 (0.84, 1.06)] (Fig 3). Apixaban was associated with similar hazards for stroke by category of TTR and increased hazards of death compared to warfarin in those with well-controlled warfarin treatment (TTR ≥ 0.75) [stroke/SE 1.05 (0.82, 1.34), all-cause death 1.20 (1.04, 1.37)] (Fig 3).

## Analysis of apixaban dose-adjustment

The proportion of patients meeting the criteria for reduced dose apixaban (Table 4) was similar between the CPRD ARISTOTLE-analogous apixaban, warfarin, and RCT apixaban groups (4.9%, 4.9%, and 4.7%, respectively). When including the additional NICE criteria of creatinine clearance, 5.1% of apixaban users in the ARISTOTLE-analogous cohort had an indication for reduced-dose apixaban, yet a larger proportion (14.3%) were prescribed reduced dose apixaban implying some patients in CPRD Aurum may have been prescribed the wrong dose and/or information on criteria for dose reduction may have been missing from CPRD Aurum.

A futher analysis of the quality of dose-adjustment in patients in CPRD Aurum (Table 5) indicated 10.5% of patients may have been prescribed an incorrect dose of apixaban at the index prescription based on the data contained in their electronic health records (EHRs). The

**Table 5. Quality of apixaban dose-adjustment in CPRD Aurum ARISTOTLE-analogous cohort.**

| Dose Status Against NICE Criteria For Dose-adjustment at Index Date | CPRD Aurum ARISTOTLE-analogous Apixaban (N = 8,846) |
|---|---|
| Patients on correct dose | 7,921 (89.5%) |
| Patients on incorrect dose | 925 (10.5%) |
| Standard 5.0 mg BID dose despite meeting criteria for dose reduction | 59 (0.7%) |
| Reduced 2.5mg BID dose despite not meeting criteria for dose reduction | 866 (9.8%) |
| 0 dose adjustment criteria recorded in EHR | 313 (3.5%) |
| 1 dose adjustment criteria recorded in EHR | 553 (6.3%) |
| Age >80 years | 389 (4.4%) |
| Body weight ≤60 kg | 57 (0.6%) |
| Serum creatinine ≥1.5 mg/dL | 107 (1.2%) |

majority of incorrect dose relating to patients being prescribed reduced-dose apixaban despite not meeting the criteria for dose reduction. A large proportion of patients prescribed an incorrect dose had only 1 dose adjustment criteria (59.6% of those with incorrect dose), suggesting some prescribers may have thought a dose reduction was warranted when only 1 criteria was present. Other possible reasons for the incorrect dose-adjustment observed here may be data on the criteria missing from the EHR (i.e., incorrect ascertainment) or consideration of other medical history that made a prescriber adjust the dose.

To assess the impact of the quality of dose-adjustment in the CPRD cohort on the effectiveness of apixaban, a supplementary post hoc analysis was performed looking at the results in the subset of apixaban patients prescribed the correct dose ($N = 7,921$) compared with IPTW rebalanced warfarin comparators. The results in this subset were consistent with the primary results showing apixaban to be noninferior to warfarin (stroke/SE 0.96 [0.78,1.17], death 0.97 [0.87,1.09]) with the results moving slightly closer to those observed in ARISTOTLE.

## Safety results

The analysis for safety outcomes is presented in Fig 4 and Table A5 in S1 Appendix; patients on apixaban had a lower risk of major bleeding compared with those on warfarin, HR (95% CI) 0.88 (0.77,1.00), consistent with ARISTOTLE. Analysis by TTR suggested superiority of apixaban for major bleeding in those with TTR <0.75 [0.74 (0.63, 0.86)], whereas apixaban

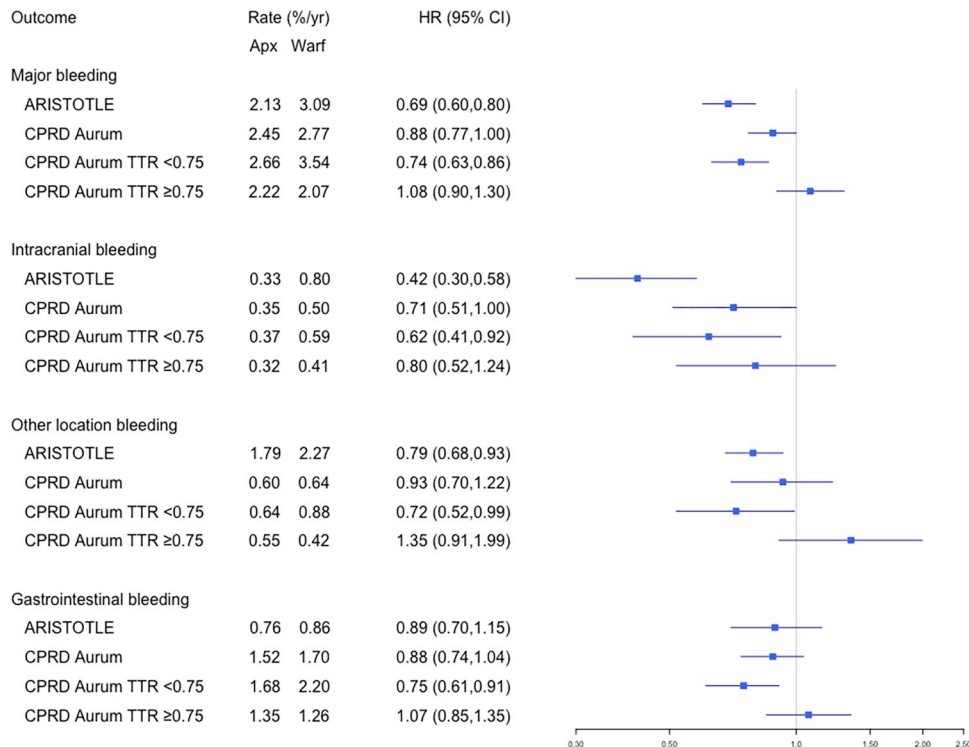

**Fig 4. Forest plot showing HRs (dots) and 95% CIs (lines) for apixaban vs. warfarin.** Absolute event rates (%/year) and HR (95% CIs) are presented for key safety outcomes in (i) ARISTOTLE, (ii) CPRD Aurum trial-matched cohort, (iii) CPRD Aurum trial-matched with TTR < 0.75, and (iv) CPRD Aurum trial-matched with TTR ≥ 0.75. For the analysis by TTR, IPTW was applied to the apixaban users targeting the treatment effect in the warfarin users with TTR <0.75 and TTR ≥0.75. CI, confidence interval; CPRD, Clinical Practice Research Datalink; HR, hazard ratio; IPTW, inverse probability treatment weighting; TTR, time in therapeutic range.

users had a similar risk of major bleeding compared with those with optimal warfarin control (TTR $\geq$ 0.75) [1.08 (0.90,1.30)].

## Sensitivity analyses

Table A7 in S1 Appendix shows the proportion of patients switching treatment. A higher proportion of patients on warfarin switched to an alternative OAC during follow-up compared with those on apixaban (16.3% versus 6.1%).

Comparing patients who switched treatment during follow-up with those that continued on index treatment (Table A8 in S1 Appendix) suggests possible selection bias due to attrition in on-treatment analyses with median TTR markedly lower in warfarin users who switched treatments compared with persistent warfarin users (median TTR 0.64 versus 0.78). On-treatment analyses would likely be biased against apixaban since patients doing badly on warfarin (i.e., with low TTR) who would be more likely to experience events in the warfarin arm would be censored at treatment switch.

On-treatment analyses censoring around treatment switch or discontinuation are presented for the effectiveness analyses in the appendix (Table A6 in S1 Appendix); the results show evidence of the expected attrition bias against apixaban when compared with the ITT results in Fig 2, for example, HR for stroke/SE is 1.04 (0.86, 1.25) in the on-treatment compared with 0.98 (95% CI 0.82, 1.19) in the ITT analysis.

Repeating the analysis with start of study period shifted forwards a year to investigate the impact of inclusion of early adopters yielded similar results to the primary analysis (Table A9 in S1 Appendix).

Prior INR control was not included in the PS models for the VKA-experienced due to a high rate of missing prior INR data (missing for 34% in the apixaban arm). A post hoc sensitivity analysis including a prior INR control variable in the PSM gave results consistent with the primary results [stroke/SE HR 95% CI 1.02 (0.86,1.21)]. Details of this post hoc analysis are in S1 Appendix.

## Discussion

In our emulation of ARISTOTLE using UK routinely collected healthcare data, we found results that met our predefined criteria for comparability with the trial. We saw noninferiority of apixaban versus warfarin for prevention of stroke or SE, all-cause mortality, and major bleeding but did not see superiority of apixaban versus warfarin for these outcomes as was seen in ARISTOTLE. We found higher TTR in the patients using warfarin in our cohort compared with the warfarin arm of ARISTOTLE (median 0.76 versus 0.66). While our analysis by TTR showed noninferiority of apixaban versus warfarin for our stroke or SE outcome, we observed an increased risk of death on apixaban compared with patients well-controlled on warfarin (TTR $\geq$ 0.75) but not when compared with those on poorly controlled warfarin (TTR < 0.75). For major bleeding, while apixaban was superior when compared to those on poorly controlled warfarin, there was no difference when compared to those on well-controlled warfarin. We saw evidence suggesting suboptimal dosing of apixaban in our cohort with approximately 10% of patients in the apixaban arm prescribed the reduced dose without meeting the criteria for the reduced dose.

We found the differences in the overall treatment-effect estimates between our cohort and ARISTOTLE may be explained by the lower proportion of Asian patients in our cohort, differences in INR control in the warfarin arm of our cohort compared with ARISTOTLE, and the higher proportion of patients prescribed a reduced dose of apixaban in our cohort compared with ARISTOTLE.

Our findings are consistent with a UK study of ischemic stroke, which compared DOACs with warfarin [29]. A Danish study found similar results to ours for stroke/SE [30], although they found apixaban users had a lower risk of death; a study of US claims data [31] also found apixaban was associated with a lower risk of death. A systematic review and meta-analysis of 16 studies [32] found pooled results for stroke and intracranial haemorrhage that were consistent with ours. One study (in US claims data) also aimed to replicate ARISTOTLE [33,34] and, in contrast to our study, found superiority for apixaban for stroke/SE, which may be linked to population differences such as lower TTR in US patients on warfarin [35] and differences in ethnicity. None of these studies matched to the ARISTOTLE trial participants, included prevalent users, or looked at how warfarin control impacted results. Further details on these studies including design and key results are summarised in Table A10 in S1 Appendix.

A key strength of our study was the use of a framework that sampled prevalent users (the continuing users of warfarin in this study) in a way that avoided selection bias facilitating the construction of a cohort of patients similar to the target trial population, which included both new users of apixaban and warfarin (VKA-naïve) and patients with prior VKA exposure (VKA-experienced) that were randomised to stay on warfarin or switch to apixaban. The use of PSM, stratified by treatment history, enabled us to select a matched cohort well balanced on important covariates. The successful emulation of ARISTOTLE by our study shows that valid treatment effects can be obtained for important outcomes with OACs using noninterventional methods with routinely collected clinical data. Having validated this framework, in future studies, we can look at the effectiveness of OACs in AF patient groups not included or underrepresented in the RCT, such as elderly patients and those at increased bleeding risk. We also recommend future analyses with an extended follow-up period compared with this study to compare the long-term outcomes seen in the noninterventional cohort with projected long-term outcomes from the RCT.

An additional strength of our study was the ability to explore the quality of warfarin treatment in our cohort and the impact of INR control on the treatment effect estimates. Our finding that the benefits of apixaban versus warfarin for some outcomes depended on the quality of INR control in the warfarin arm answers questions raised in the NICE premeeting briefing, which looked at apixaban in the NVAF population and noted the TTR seen in ARISTOTLE "may be lower than what is typical in UK clinical practice" and "apixaban compared with well-controlled warfarin (TTR 75% or more) may not be superior in the long term" [8]. ARISTOTLE presented outcomes by centre (for example, hospital) TTR quartile and did not show a signal of treatment efficacy differing by centre TTR quartile. We were able to use IPTW to estimate the treatment effect in the different warfarin TTR groups and used predicted TTR for warfarin users missing TTR to attempt to limit the risk of selection bias.

While our study aimed to emulate ARISTOTLE using suitable methods, there were several limitations. Some of the criteria assessed for ARISTOTLE eligibility may not be well recorded in CPRD leading to a risk of misclassification. Furthermore, misclassification of ARISTOTLE eligibility criteria and baseline covariates could be differential by treatment in the VKA-experienced patients if criteria such as renal function are more likely to be checked before changing treatment. However, the most important risk factors for the primary outcome of stroke (the components of $CHA_2DS_2$-VASc stroke risk score) are mostly well recorded in CPRD Aurum and HES.

Our cohort did not attempt to match the trial on the use of concomitant medications in order for our cohort to reflect typical UK prescribing. In ARISTOTLE, 31% of participants were using aspirin and 11% using amiodarone at baseline, whereas in our cohort, only 6% were recorded as using aspirin and 4% amiodarone. Amiodarone potentiates the effects of warfarin, and concomitant use of amiodarone with DOACs is associated with increased risk of major bleeding [36], while concomitant use of aspirin increases the risk of bleeding for both warfarin [37] and DOACs

[38]. The difference in concomitant medication usage between our cohort and the trial population may explain some of the observed differences in treatment effects.

A key limitation of our study was the inability to match ARISOTLE on ethnicity, meaning the CPRD Aurum cohort included a low number of patients from Asian and Hispanic groups when compared with the RCT (14.5% of participants in ARISTOTLE were Asian compared to 2.4% in our ARISTOTLE-analogous CPRD cohort). There are known racial differences in the treatment effects of OACs with Asian patients experiencing a higher risk of haemorrhagic stroke and intracranial haemorrhage compared with white patients; in ARISTOTLE, Asian participants experienced double the risk of stroke or SE when on warfarin therapy when compared with white participants [39]. The reasons for the increased risk of bleeding associated with warfarin therapy in Asian patients is hypothesised to be associated with differences in drug metabolism and prevalence of cerebral microbleeds [40]. The difference in proportion of Asian patients between our cohort and ARISTOTLE is therefore likely to explain some of the differences in treatment effects seen and limits the generalisability of our study, with the results of our study of most relevanance to white patients. This limitation on ethnicity arose from the data source used and time period studied (patients with AF in CPRD Aurum 2013–2019), which had a low proportion of Asian patients, likely due to AF being associated with older age combined with a lower prevalence of AF in Asian patients compared with white patients [41]. While CPRD Aurum is largely representative of the UK population in relation to ethnicity [42], diversity is still limited for older individuals. Despite this, CPRD Aurum has shown to be a useful resource for investigating treatment effects in different ethnic groups for indications such as hypertension, which is more prevalent and occurs at a younger age in ethnic minority groups, with similar trial replication methods used to compare antihypertensive treatment effects in underrepresented ethnic groups [13].

The approach our study used for handling missing data on baseline covariates relied on assumptions on the relationship between missingness, treatment, and outcomes, which may not be valid; however, the low proportion of missing data means that this is unlikely to have impacted the results. In the coarsened exact matching step, the choice of variables will have an impact on the resulting cohort selected, meaning a different combination of variables could lead to different results. There is a risk that residual confounding may be present despite the use of PSM. The use of PSM also has the potential to introduce bias by dropping patients from the cohort [19]; however, PSM is well suited to the process of trial emulation including prevalent users, and a low number of apixaban users were dropped due to unsuccessful matching. The inclusion of prevalent users of warfarin in the cohort risks the introduction of selection bias [20,21]; this was avoided by use of a method shown to produce unbiased estimates in a simulation study [21]. We found consistent results between our new and prevalent user strata across multiple outcomes providing reassurance the method used was likely to have successfully avoided selection bias.

Apixaban along with other DOACs were rapidly adopted as preferred first-line OAC in AF during the study period; it was therefore not possible to match on calendar date leading to a difference in follow-up time between the treatment arms in our cohort. A higher proportion of warfarin users switched to alternative OAC during follow-up compared with those prescribed apixaban (16% versus 6%). The impact of this differential switching during follow-up was addressed in the sensitivity analyses. The availability of new alternative treatments during the study period also means there is a risk of channelling bias in that over time the patients still on warfarin are more likely to be those doing well on warfarin. INR control prior to the index date was not included in the PS for the prevalent users due to a high rate of missing data; however, other variables associated with poor INR control were included in the models, and an exploratory post hoc analysis including a variable for poor INR control gave results consistent with the primary results.

Adherence to treatment was difficult to assess in our study due to automatic repeat prescriptions; treatment persistence was more useful in providing a measure of pattern of medicine use over time. In the analysis by TTR, the adherence of patients using apixaban was not accounted for; however, a previous UK study showed apixaban had higher adherence than VKAs [43], meaning we would expect to see better effectiveness outcomes in apixaban. Futhermore, the use of IPTW in the analysis by TTR means predictors of poor adherence are likely to have been balanced between treatments. The analysis of TTR is limited by this being a post-baseline measure available for only 1 treatment arm leading to a risk of selection bias in this analysis—patients with TTR available in the study may be more healthy than those without this measure given that patients have to survive and not be hospitalised to have INR measurements available in CPRD Aurum. The limitation of use of a post-baseline measurement available for 1 treatment arm was also evident in the RCTs of DOACs versus warfarin and is mitigated in our study through the use of IPTW and predicted TTR for patients that were missing TTR (using a model to predict TTR that used INR measurements restricted to the first year of follow-up). Given the risk of selection bias in the analysis by TTR and risk of misclassification of TTR for those missing TTR, these results should be considered exploratory and interpreted with caution. Sensitivity analyses in our cohort using an on-treatment censoring scheme showed evidence of attrition bias. The regular measurement of INR and availability of alternative anticoagulants makes warfarin therapy particularly prone to attrition bias since a patient may be more likely to switch to a DOAC if their INR is frequently out of the optimal range or if they have not been adhering to scheduled INR testing.

To conclude, we found that applying a reference trial emulation approach allowed us to emulate a landmark randomised trial of apixaban versus warfarin using UK noninterventional data, with results meeting prespecified benchmarking criteria based on the reference trial results. This trial emulation method provides valid treatment effect estimates for apixaban compared to warfarin and can be used to determine risks and benefits of AF medications in people treated in routine clinical care. This study demonstrates a successful real-world application of novel methods that have been proposed for the inclusion of prevalent users in observational studies, with the application of an adaptation to mimic the screening process making the method suitable for emulation of RCTs that include prevalent users. These methods could be adapted for emulation of RCTs in other therapeutic areas and for looking at patient groups underrepresented or excluded from RCTs.

The weaker overall treatment benefit observed in our cohort appears to be due to a higher proportion of patients with well-controlled warfarin in the UK clinical context, compared with the trial. Our exploratory analysis by TTR showed similar results for stroke and a greater benefit for apixaban for major bleeding compared with TTR <0.75; conversely, a slightly higher risk of death was observed on apixaban compared with well-controlled warfarin.

## Disclaimer

The views expressed in this paper are those of the author and not do not necessarily reflect those of the SFDA or its stakeholders. Guaranteeing the accuracy and the validity of the data is a sole responsibility of the research team.

## Supporting information

**S1 STROBE Checklist. Checklist of items that should be included in reports of cohort studies.**
(DOCX)

**S1 ISAC Protocol. ISAC Protocol for the ARISTOTLE emulation study.**
(PDF)

**S1 Appendix. Containing supporting information.** Table A1. ARISTOTLE inclusion and exclusion criteria applied to CPRD Aurum. Table A2. Efficacy outcomes results from ARIS-TOTLE. Table A3. Effectiveness outcomes results in the CPRD Aurum ARISTOTLE-analogous cohort. Table A4. Bleeding outcomes and net clinical outcomes results from ARISTOTLE RCT. Table A5. Bleeding outcomes and net clinical outcomes results in the CPRD Aurum ARISTOTLE-analogous cohort. Table A6. Effectiveness outcomes results in the CPRD Aurum ARISTOTLE-analogous cohort using the on-treatment censoring scheme. Table A7. Treatment status of apixaban and warfarin users in CPRD Aurum ARISTOTLE-analogous cohort during 2.5 years of follow-up. Table A8. Characteristics of apixaban and warfarin users in CPRD Aurum ARISTOTLE-analogous cohort by treatment persistence during 2.5 years of follow-up. Table A9. Effectiveness outcomes results in the CPRD Aurum ARISTOTLE-analogous cohort using later study start date (1 January 2014). Table A10. Summary of noninterventional studies comparing apixaban and warfarin in AF patients.
(DOCX)

## Author Contributions

**Conceptualization:** Emma Maud Powell, Usha Gungabissoon, Ian J. Douglas, Kevin Wing.

**Data curation:** Emma Maud Powell, Turki M. Bin Hammad.

**Formal analysis:** Emma Maud Powell.

**Investigation:** Emma Maud Powell.

**Methodology:** Emma Maud Powell, Usha Gungabissoon, John Tazare, Liam Smeeth, Paris J. Baptiste, Turki M. Bin Hammad, Angel Y. S. Wong, Ian J. Douglas, Kevin Wing.

**Project administration:** Emma Maud Powell.

**Supervision:** Usha Gungabissoon, Ian J. Douglas, Kevin Wing.

**Writing – original draft:** Emma Maud Powell.

**Writing – review & editing:** Emma Maud Powell, Usha Gungabissoon, John Tazare, Liam Smeeth, Paris J. Baptiste, Turki M. Bin Hammad, Angel Y. S. Wong, Ian J. Douglas, Kevin Wing.

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
