## [Editor Report · Decision Letter 0]

19 Sep 2023

Dear Dr Powell, 

Thank you for submitting your manuscript entitled "Comparative treatment effects of oral anticoagulants for stroke prevention in atrial fibrillation: analysis of routinely collected data with validation against results from a randomised controlled trial" for consideration by PLOS Medicine.

Your manuscript has now been evaluated by the PLOS Medicine editorial staff as well as by an academic editor with relevant expertise and I am writing to let you know that we would like to send your submission out for external peer review.

Please re-submit your manuscript within two working days, i.e. by Sep 21 2023 11:59PM.

Kind regards,

Pippa

Philippa Dodd, MBBS MRCP PhD

PLOS Medicine

---

## [Decision Letter · Decision Letter 1]

6 Nov 2023

Dear Dr. Powell,

Thank you very much for submitting your manuscript "Comparative treatment effects of oral anticoagulants for stroke prevention in atrial fibrillation: analysis of routinely collected data with validation against results from a randomised controlled trial" (PMEDICINE-D-23-02702R1) for consideration at PLOS Medicine. 

[LINK]

In light of these reviews, I am pleased to tell you that we would like to consider a revised version that addresses the reviewers' and editors' comments. Obviously we cannot make any decision about publication until we have seen the revised manuscript and your response, and we plan to seek re-review by one or more of the reviewers. 

We expect to receive your revised manuscript by Nov 27 2023 11:59PM. Please email us (plosmedicine@plos.org) if you have any questions or concerns.

We look forward to receiving your revised manuscript. PLease don't hesitate to contact me via the email address detailed below if you have any questions or concerns.

Best wishes,

Pippa

Philippa Dodd, MBBS MRCP PhD

PLOS Medicine

plosmedicine.org

pdodd@plos.org

COMMENTS FROM THE ACADEMIC EDITOR

To be able to use routine clinical data from an unselected population in a way that methodologically correctly simulates a randomized trial is very important and useful whenever a new treatment is introduced through industry-sponsored trials only. Actually, even when the trial is performed independently, the issue of a selected population still remains, and results from such an analysis in routinely collected data will still be very worthwhile regarding generalizability.

This paper therefore has a double message that is worthwhile from both respects: the clinical message that in fact warfarin is not inferior to apixaban if the warfarin treatment is good; and the scientific message that it would be good to perform such studies whenever new treatments are being studied.

Having said that, there is maybe one limitation that is not addressed in detail, which is the issue of residual confounding. Even properly performed propensity scores cannot take away unmeasured confounding. The fact that a strong effect is found for death (HR 0.84) for the subjects with TTR in the low range suggests in my opinion that some confounding may still be present, as low TTR is generally associated with worse health and more comorbidity. Also the fact that the effect estimates flip so clearly from below to above one suggests this phenomenon. But even if residual confounding would still be present, I don’t think that would interfere with the conclusion.

COMMENTS FROM THE EDITORS

GENERAL

Please respond to all editor and reviewer comments detailed below in full.

Please include line numbers starting at 1 on the title page and in continuous sequence throughout, thereafter.

Please ensure that the study is reported according to the STROBE guideline, and include the completed STROBE checklist as Supporting Information. Please add the following statement, or similar, to the Methods: "This study is reported as per the Strengthening the Reporting of Observational Studies in Epidemiology (STROBE) guideline (S1 Checklist)."

When completing the checklist, please use section and paragraph numbers, rather than page numbers, as these often change in the event of publication.

TITLE

In your abstract ‘objective’ you state, ‘To validate non-interventional methodology…’. However, the current title really only reflects the subject matter that you use to ‘showcase’ your methodology. We suggest revising your title to better reflect/include the former, according to PLOS Medicine’s style. 

Your title must be nondeclarative and not a question. It should begin with main concept if possible. "Effect of" should be used only if causality can be inferred, i.e., for an RCT. Please place the study design ("A randomized controlled trial," "A retrospective study," "A modelling study," etc.) in the subtitle (ie, after a colon).

Perhaps, ‘Comparative analysis of routinely collected data against results from a randomised controlled trial of oral anticoagulants for stroke prevention in atrial fibrillation: A validation study of non-interventional methodology’ or similar?

ABSTRACT

Please structure your abstract using the PLOS Medicine headings (Background, Methods and Findings, Conclusions).

Please combine the Methods and Findings sections into one section, “Methods and findings”

Abstract Background: Provide the context of why the study is important. The final sentence should clearly state the study question.

Abstract Methods and Findings:

Please ensure that all numbers presented in the abstract are present and identical to numbers presented in the main manuscript text.

Please include the study design, population and setting, number of participants, years during which the study took place, length of follow up, and main outcome measures.

Please define statistical abbreviations at first use for the reader – HR, 95% CI for example.

Please quantify the main results with 95% CIs and p values.

Please include any important dependent variables that are adjusted for in the analyses.

Please include the actual amounts and/or absolute risk(s) of relevant outcomes (including NNT or NNH where appropriate), not just relative risks or correlation coefficients. (example for absolute risks: PMID: 28399126). 

In the last sentence of the Abstract Methods and Findings section, please describe the main limitation(s) of the study's methodology.

AUTHOR SUMMARY

At this stage, we ask that you include a short, non-technical Author Summary of your research to make findings accessible to a wide audience that includes both scientists and non-scientists. The authors summary should consist of 2-3 succinct bullet points under each of the following headings:

• Why Was This Study Done? Authors should reflect on what was known about the topic before the research was published and why the research was needed.

• What Did the Researchers Do and Find? Authors should briefly describe the study design that was used and the study’s major findings. Do include the headline numbers from the study, such as the sample size and key findings. 

• What Do These Findings Mean? Authors should reflect on the new knowledge generated by the research and the implications for practice, research, policy, or public health. Authors should also consider how the interpretation of the study’s findings may be affected by the study limitations. In the final bullet point of ‘What Do These Findings Mean?’, please describe the main limitations of the study in non-technical language.

Author Summary should immediately follow the Abstract in your revised manuscript. This text is subject to editorial change and should be distinct from the scientific abstract. Please see our author guidelines for more information: https://journals.plos.org/plosmedicine/s/revising-your-manuscript#loc-author-summary

METHODS and RESULTS

Page 4 – ‘…attack (TIA)/SE…’ please define ‘SE’ here for the reader.

Page 5, line 11 – please define ‘PS’ as ‘Propensity Score (PS) model (above table 1 title).

Page 5, final paragraph – please revise to ‘threshold of 0.05’

Page 7 para 2 – it might be helpful to redefine ‘SE’ here.

Page 7, statistical analysis – thank you for indicating that your study had a prospective protocol/analysis plan. Please include the relevant prospectively written document with your revised manuscript as a Supporting Information file to be published alongside your study, and cite it in the Methods section. A legend for this file should be included at the end of your manuscript. 

Thank you for detailing the changes made to your analysis in the appendix. Please detail the nature of the changes made to your analysis, including any made in response to peer review, in the Methods section of the paper with rationale.

For all observational studies, in the manuscript text, please indicate: (1) the specific hypotheses you intended to test, (2) the analytical methods by which you planned to test them, (3) the analyses you actually performed, and (4) when reported analyses differ from those that were planned, transparent explanations for differences that affect the reliability of the study's results. If a reported analysis was performed based on an interesting but unanticipated pattern in the data, please be clear that the analysis was data-driven.

We appreciate your effort to ensure that your manuscript is succinct, but the methods section is currently rather brief, relying on the reader to refer to different pieces of documentation for nuance. Please provide additional detail of the databases you use and of the ARISTOTLE trial itself in the main manuscript.

Throughout the main manuscript where you report statistical information, please ensure to define all statistical abbreviations for the reader at first use. When reporting 95% CIs please use commas to separate upper and lower bounds as hyphens can be confused with reporting of negative values. Please also report p values where reporting 95% CIs. 

Page 8, para 2 – please quantify percentages with numerators and denominators.

For example, page 10, para 3 should read– ‘[HR 0.79 (95% CI 0.66,0.95); p</=]’. Please check and amend throughout all sections of the main manuscript and supporting files to ensure accuracy in reporting. Note that later you also use the word ‘to’.

Page 10, final para – please use an alternative to ‘…AF patients…’ perhaps, ‘…patients with AF on…’ or similar. Please check and amend throughout all subsections of the manuscript, tables, figures and supporting files where relevant.

TABLES

Throughout when reporting p values please report as <0.001 and where higher the exact p value as 0.002, for example. Please amend throughout all sections of the main manuscript and appendix where relevant.

Throughout, when using abbreviations please ensure that all are clearly defined for the reader in an appropriate footnote/caption, including those used for statistical reporting. For example, table 1 – AF, CHADS2, TIA and others are outstanding and table 2 – CI, TTR. Please check and amend throughout all tables in the main manuscript and supporting information.

FIGURES

Throughout, please ensure to define all abbreviations within an appropriate footnote, including those used for statistical reporting. For example, figure 1, ‘VKA, RCT, PSM’ and ‘figure 2, ‘CPRD, TTR’.

Throughout, when reporting 95% CIs please use commas instead of hyphens to separate upper and lower bounds as the latter can be confused with reporting of negative values.

Throughout, where 95% CIs are reported please also report p values. As above, please report p as <0.001 and where higher the exact p value as 0.002, for example. Please check and amend throughout all sections of the main manuscript and supporting files.

Please ensure your clearly define (in the figure captions) the meaning of the dots and lines used in the forest plots.

DISCUSSION

Thank you for structuring your discussion according to PLOS Medicine’s style. Please remove all sub-headings from the discussion such that it reads as continuous prose.

The opening sentence of the discussion is very long and as such rather inaccessible to the reader. Please revise for brevity and clarity summarizing the main findings of your study. It would be helpful to clearly and explicitly state how and where your data outcomes differ and align with those of the ARISTOTLE trial.

Please remove the conflict of interest, funding and data availability statements from the end of the main manuscript and include only in the manuscript submission form. In the event of publication, they will be compiled as metadata.

SUPPORTING INFORMATION

Please ensure that all tables and figures in the supporting information follow the same guidance as detailed above for the main manuscript.

Please cite your Supporting Information as outlined here: https://journals.plos.org/plosmedicine/s/supporting-information

In the published article, supporting information files are accessed only through a hyperlink attached to the captions. For this reason, you must list captions at the end of your manuscript file. You may include a caption within the supporting information file itself, as long as that caption is also provided in the manuscript file. Do not submit a separate caption file.

As above, please include the completed STROBE checklist as Supporting Information. When completing the checklist, please use section and paragraph numbers, rather than page numbers as the latter often change in the event of publication.

REFERENCES

In the bibliography, please ensure that you list up to but no more than 6 author names followed by et al.

For all web references please ensure you include an, ‘Accessed [date].’

Journal name abbreviations should be those listed in the National Center for Biotechnology Information (NCBI) databases.

Please see our website for other reference guidelines https://journals.plos.org/plosmedicine/s/submission-guidelines#loc-references

COMMENTS FROM THE REVIEWERS:

Reviewer #1: 1. Thank your your hard work. It's well-written and well-cited report. However, I am not convinced that these analyses add anything new. The subanalysis of "special" groups e.g. patients with cancer, obesity etc. may be more interesting as data are limited.

2. There are known racial differences in the treatment effect of OACs. In your study, 95.7% of patients were White, but in the Aristotle 14.4% of patients were Asians. My concern is the generalizability of the study; and it is worth to discuss as a study limitation. 

3. Do you have the data on Apixaban doses? Were they well-adjusted. It is also worth mentioning the higher number of patients with CKD (comparing to the Aristotle), which may affect dosing. 

Reviewer #2: Alex McConnachie, Statistical Review

The paper by Powell and colleagues uses CPRD data to perform propensity score matched analyses designed to replicate the ARISTOTLE trial of apixaban vs. warfarin for the treatment of AF. This review looks at their use of statistics.

This is an expertly designed and delivered analysis. The construction of the study cohort is very cleverly done, using various steps to ensure comparability, first to the ARISTOTLE population (where possible) and then between exposed and unexposed individuals. The analysis methods themselves are all good. The Cox models use robust standard errors to account for the matched design, and are appropriately checked for proportional hazards. Sensitivity analyses are an on-treatment analysis and an analysis using a later start date.

Subgroup analyses are done by TTR. Is this just for the patients who switched from warfarin to apixaban, with TTR relating to the period prior to the index date? I note that TTR is not included in Table 2. Also, TTR does not appear to be included in Table 1, as one of the covariates used in the PS matching procedure - should it be? If channelling is a risk, then surely patients with worse TTR might be more likely to be switched to apixaban?

Another possibility is that the subgroup analysis is performed in relation to the TTR of the warfarin users after the index date. After multiple reads of the paper, I believe this may be what was done. Is it appropriate to stratify by a post-baseline characteristic?

For any subgroup analysis I would normally ask to see an interaction test p-value, but that may not be appropriate here (or necessary, given the magnitude of the difference in treatment effect estimates).

The wording of the main conclusion in the abstract did not seem to match the results on first reading. Yes, the results showed non-inferiority, as did ARISTOTLE, but clearly not superiority, which ARISTOTLE did. To say the results are comparable therefore seems incongruous. To describe the treatment effect observed as "weaker" doesn't seem to fit. Looking at the effect estimates and CIs, I would describe the results as showing no treatment effect in the full population. The observation that this is a combination of a treatment effect benefit in the TTR<0.75 group and a treatment harm in the TTR>=0.75 group then makes sense.

Finally, and I don't know enough about the theory to be definitive on this, but I have read that propensity score matching can introduce bias rather than remove it. What assurances can the authors give that the way they have implemented PSM is good? Does it come down to the choice of caliper width?

Reviewer #3: Summary

The authors examined the impact of warfarin time in therapeutic range on outcomes using ARISTOTLE-analogous cohort in the UK. The research concepts are intriguing, the study findings appear highly significant, and the study was well-written.

Here are some comments for revision to improve the manuscript:

Comments

1. Introduction: It is recommended to provide a more detailed explanation of how the results would remain consistent or similar to RCT results, even when the population that was excluded from or underrepresented in trials is included. In this regard, the 'validation of results against ARISTOTLE' section in the Methods section may need further elaboration for the benefit of readers.

2. Methods: For the definition of a new user of warfarin or a warfarin-naïve apixaban user, is there a required washout period to ensure their non-use of warfarin for a certain duration?

3. Methods: Why was a caliper of 0.2 chosen for use?

4. Results: 354 apixaban patients who had no matches were mentioned, along with 8846 patients who remained after matching. The numbers do not align (9120 - 354 = 8766).

5. Discussion: In an RCT, participants are expected to maintain good adherence to their assigned treatment. However, in this study, I couldn't find a clear operational definition or justification for 'persistent patients' (e.g., checks at every 6-month time point would suffice?), 'discontinuation' (e.g., a definition using the number of days between prescriptions?), and 'switching' (e.g., when a prescription for a switching drug occurred during or certain days after the index treatment's prescription, how should this be considered?). It would be beneficial to provide further elaboration regarding these in both the Methods and Discussion sections.

6. Discussion: Is emulating an RCT in 2014 still relevant for today's real-world patient populations, given the changing preferences of patients and physicians, as well as evolving clinical practices, guidelines, and recommendations?

7. Discussion: Could we explore future analyses with an extended follow-up period stemming from this study, particularly for the purpose of projecting long-term outcomes from an RCT? Are there any other valuable directions or extensions for this study that we should consider?

[LINK]

---

## [Decision Letter · Decision Letter 2]

8 Mar 2024

Dear Dr. Powell,

Thank you very much for re-submitting your manuscript "Comparison of oral anticoagulants for stroke prevention in atrial fibrillation: a cohort study in the UK Clinical Practice Research Datalink with emulation of a reference trial (ARISTOTLE)" (PMEDICINE-D-23-02702R2) for review by PLOS Medicine.

I have discussed the paper with my colleagues and it was also seen again by the academic editor and the statistical reviewer. I am pleased to say that provided the remaining editorial and production issues are dealt with we are planning to accept the paper for publication in the journal.

[LINK]

We look forward to receiving the revised manuscript by Mar 15 2024 11:59PM.   

Kind regards,

Pippa

Philippa Dodd, MBBS MRCP PhD

PLOS Medicine

plosmedicine.org

pdodd@plos.org

Requests from Editors:

GENERAL

Thank you for your detailed and considered responses to previous editor and reviewer comments. Please see below for further comments which we require that you address prior to publication.

TITLE

Suggest emphasizing more clearly the fact that this is target trial emulation study. Perhaps, 

“Comparison of oral anticoagulants for stroke prevention in atrial fibrillation using the UK Clinical Practice Research Datalink Aurum: A reference trial (ARISTOTLE) emulation study”. 

ABSTRACT

Abstract Background:

We appreciate the inclusion of additional detail which is very helpful to ascertain the aim of your study. For the purpose of brevity suggest removing lines 25-27 from the abstract and restricting to the introduction of main manuscript. Suggesting including a final sentence in the 1st paragraph of ‘Background’ to read as follows:

‘Reference trial emulation allows evidence to be generated around treatment effects in groups excluded or underrepresented in the original trials.’

Abstract Methods & Findings:

Line 37 – suggest revising ‘warfarin CPRD users’ to ‘warfarin users in the CPRD Aurum’.

Abstract Conclusions:

Do these data tell us anything about the usefulness of this methodological approach to investigate treatment effects for other conditions? 

AUTHOR SUMMARY

Line 65 – suggest, ‘This study used routinely collected health data from the UK to…’

Lines 69-70 – suggest, ‘…help to understand how transferrable RCT results are to ‘real-world’ practices and whether this methodological approach can help to improve treatment options and outcomes for patient groups currently underrepresented in clinical trials.’ Or similar.

Line 75 – suggest, ‘UK primary care data’.

Line 81 – sentence beginning ‘This may be explained…’ suggest removing and (after paraphrasing) suggest placing as part of the final bullet point of the ‘What do these findings mean?’ sub-section as a limitation.

Lines 88 and 90 – suggest swapping these 2 bullet points around such that the ‘benefits’ of the methodological approach are listed sequentially.

Line 93 – suggest removing the word ‘these’ as this methodology could be applied to various treatment strategies for a number of different conditions – this would be worth emphasizing as it is a major selling point of your study, at least in my opinion! Might also be worth giving an example and making a clear distinction between those that are excluded from clinical trials (those with multimorbidity, for example) and those who are underrepresented (certain ethnic minority groups, for example).

Line 96 – please revise the final point in line with previous comments (see line 81).

INTRODUCTION

As noted above, the methodological approach is a really big selling point which deserves emphasis but not only for anticoag. in AF, also for a variety of treatments and differing conditions.

As above, might be worth differentiating & defining the excluded and underrepresented groups so that readers can make tangible associations, which should increase the impact of your take-home message.

Line 140 – would ‘influenced by’ be more accurate than ‘dependent upon’?

TABLES and FIGURES

Table 3 – for the purpose of formatting requirements, please include a leading zero for all numerical values in the standardized difference column, for example, row 3 should read ‘0.008’. If not for the purpose of transparent reporting, please clearly explain the reasons why not.

DISCUSSION

Some parts of your discussion read very well and others are less nuanced and focused. Suggest revising for improved nuance and clarity. The strengths and limitations of your findings in context of your chosen methodological approach, which we think has clear strengths and transferability as well as some limitations, could be more clearly presented in parts. 

Please remove all sub-headings such that the discussion reads as continuous prose.

Lines 528-530 – when referring to ‘prevalent users’ do you mean users of warfarin, apixaban or both. Please amend for clarity.

Line 568 onwards – you discuss here the implications of ethnicity on your results compared to ARISTOTLE. For further clarity, could you elaborate on the reasons why such a difference was observed. Was this a consequence of the database or an effect of PSM? Considering that this (methodological) approach could help in ascertaining treatment effects in those who are underrepresented in trials (such as ethnic minority groups), what implications might this finding have on the usefulness of either the dataset or the methodological approach more broadly? 

SUPPORTING INFORMATION

Please ensure that abbreviations are defined throughout the supporting information, including tables and figures as relevant.

Thank you for including the published protocol. Apologies for the confusion at the time of our previous request we were asking for the original protocol document (i.e., as used for your study approval process or data access process) not the BMJ published protocol article, this does not need to be included, the reference will suffice. Please amend.

SOCIAL MEDIA

To help us extend the reach of your research, please detail any X (formerly Twitter) handles you wish to be included when we tweet this paper (including your own, your coauthors’, your institution, funder, or lab) in the manuscript submission form when you re-submit the manuscript.

Comments from Reviewers:

Reviewer #2: Alex McConnachie, Statistical Review

I thank the authors for their consideration of my original comments. I am happy with their responses, and have no further comments to make.

[LINK]

---

## [Editor Report · Decision Letter 3]

23 May 2024

Dear Dr. Powell,

Thank you very much for re-submitting your manuscript "Comparison of oral anticoagulants for stroke prevention in atrial fibrillation using the UK Clinical Practice Research Datalink Aurum: A reference trial (ARISTOTLE) emulation study" (PMEDICINE-D-23-02702R3) for review by PLOS Medicine.

As discussed via email please re-submit your manuscript once the required revisions to the analyses are complete.

We look forward to receiving the revised manuscript by 6th June but if you require more time then please let me know. 

Kind regards

Pippa

Philippa Dodd, MBBS MRCP PhD 

PLOS Medicine

plosmedicine.org

pdodd@plos.org

[LINK]

---

## [Decision Letter · Decision Letter 4]

12 Jun 2024

Dear Dr Powell, 

On behalf of my colleagues and the Academic Editor, Professor Suzanne Cannegieter, I am pleased to inform you that we have agreed to publish your manuscript "Comparison of oral anticoagulants for stroke prevention in atrial fibrillation using the UK Clinical Practice Research Datalink Aurum: A reference trial (ARISTOTLE) emulation study" (PMEDICINE-D-23-02702R4) in PLOS Medicine.

PRESS

Kind regards,

Pippa 

Philippa Dodd, MBBS MRCP PhD 

Senior Editor 

PLOS Medicine

pdodd@plos.org